# Calcitriol in the Presence of Conditioned Media from Metastatic Breast Cancer Cells Enhances Ex Vivo Polarization of M2 Alternative Murine Bone Marrow-Derived Macrophages

**DOI:** 10.3390/cancers12113485

**Published:** 2020-11-23

**Authors:** Artur Anisiewicz, Natalia Łabędź, Izabela Krauze, Joanna Wietrzyk

**Affiliations:** Department of Experimental Oncology, Hirszfeld Institute of Immunology and Experimental Therapy, Polish Academy of Sciences, 53-114 Wroclaw, Poland; natalia.labedz@hirszfeld.pl (N.Ł.); izakrauze_8@interia.pl (I.K.); joanna.wietrzyk@hirszfeld.pl (J.W.)

**Keywords:** macrophages, BMDMs, vitamin D, calcitriol, breast cancer cells, M2 macrophages, 4T1 cells, 67NR cells

## Abstract

**Simple Summary:**

In this study, we stimulated bone marrow-derived macrophages to M0, M1, and M2 subtypes, with or without calcitriol, or with or without 4T1 (metastatic), 67NR (non-metastatic), and Eph4-Ev (normal) cell culture supernatants (CMs) to test their effect on polarization. We showed that calcitriol increased the expression of *Cd206* and *Spp1* mRNA and CD36, CCL2, and arginase levels for M2 macrophages and decreased *Cd80* and *Spp1* mRNA and IL-1, IL-6, OPN, and iNOS for M1 macrophages. 4T1 CM influenced the expression of the studied genes and proteins to a greater extent than 67NR and Eph4; the strongest effect was noted for M2 macrophages. We show that calcitriol and 4T1 CM enhance the polarization of M2 macrophages and M2 macrophages differentiated with calcitriol-stimulated migration of 4T1 and 67NR cells. We indicate that the immunosuppressive properties of calcitriol may unfavorably affect the tumor microenvironment, and supplementation with vitamin D in oncological patients may not always bring benefits.

**Abstract:**

In this study, we differentiated murine bone marrow-derived macrophages (BMDMs) into M0, M1, and M2 in the presence or absence of calcitriol. Real-time PCR analysis of gene expression, FACS analysis of surface markers, and chemokine/cytokine production assays were performed. In addition, the effect of the conditioned media (CM) from murine breast cancer 4T1 (metastatic) and 67NR (non-metastatic) and Eph4-Ev (normal) cells with and without calcitriol on the polarization of M1/M2 cells was determined. We found that calcitriol enhanced the differentiation of M2 macrophages, which was manifested by increased expression of *Cd206* and *Spp1* mRNA and CD36, Arg, and CCL2 in M2 BMDMs and by decreased expression of *Cd80* and *Spp1* mRNA and IL-1, IL-6, OPN, and iNOS in M1 BMDMs. 4T1 CM showed a higher effect on the gene and protein expression in macrophages than 67NR and Eph4-Ev, with the greatest effect observed on M2 macrophages which increased their differentiation and properties characteristic of alternative macrophages. Moreover, M2 macrophages differentiated with calcitriol-stimulated migration of 4T1 and 67NR cells through fibronectin and collagen type IV, respectively. Overall, our results indicated that vitamin D supplementation may not always be beneficial, especially in relation to cancers causing excessive, pathological activation of the immune system.

## 1. Introduction

Macrophages are a key element of innate immunity. Due to their numerous functions in all tissues, these cells are essential for the maintenance of local and overall homeostasis [1]. The phenotype of macrophages is highly plastic and determined by the occurrence of specific stimulating factors. Under the influence of factors released by cancer cells, such as chemokine CCL2, peripheral monocytes and local macrophages are recruited to the primary tumor and transformed into tumor-associated macrophages (TAMs) [2,3]. Although M1 classical macrophages exhibit antitumor activity, during cancer progression, a predominance of immunosuppressive macrophages is generally observed. These macrophages, which are phenotypically similar to M2 alternative macrophages, support the growth of primary tumors, increase the metastatic potential of tumor cells, and promote vascularization and remodeling of tumor stroma, thus accelerating cancer progression [4,5,6]. In many cancers, including breast cancer, a high rate of TAMs infiltration is basically associated with a poor prognosis [6,7]. It was also reported that overall survival or disease-free survival mostly correlated with the expression of the M2 phenotype: CD163+, CD204+, or CD206+ [8,9].

Calcitriol (1,25-dihydroxyvitamin D_3_), the biologically active form of vitamin D, is a molecule exhibiting pleiotropic effects, including those affecting the immune system. Several authors have described the effects of calcitriol, using different models of monocytes/macrophages, including: (1) reduction in the expression of pro-inflammatory cytokines such as IL-12 [10], IL-6, TNF-α [11,12], IL-1 [13], IL-23, GM-CSF [14], and iNOS [15]; (2) decrease in the expression of CD80 and CD86 [13] and ability to present antigens (decrease in MHC II) [14]; (3) increase in the expression of antimicrobial peptides, namely defensin and cathelicidin [16]; and (4) increase in the expression of IL-10 [17], ARG1 [18,19], CD163, and CD206 [18]. However, the effect of calcitriol on the TAMs phenotype is still unknown.

The antitumor properties of calcitriol have been well established [20,21,22,23]. However, our previous studies showed that calcitriol and its hypocalcemic analogs stimulated the metastasis of 4T1 mammary gland carcinoma in BALB/c mice [24]. Further, in the case of non-metastatic 67NR mouse mammary gland cancer, it turns out that a high dietary cholecalciferol content causes the appearance of cancerous cells in the lungs [25]. In addition, after calcitriol treatment, stimulation of metastatic spread was noted in the TRAMP mouse prostate cancer model [26], while Cao et al. reported the stimulation of tumor growth after vitamin D supplementation in the 4T1 model [27]. Our further studies on the effects of calcitriol and its analogs on immunity in 4T1 tumor-bearing mice showed an enhancement of Th2 response and Treg lymphocyte activity [28], and stimulation of Th17 cell differentiation [29]. Moreover, we found that increased metastasis was accompanied by an increase in the percentage of immunosuppressive Ly6C^low^ splenic monocytes and an elevated concentration of cytokines and proteins that are characteristic of M2 macrophages in the tumor [30]. Moreover, the results of clinical trials are also inconclusive. A 2018 meta-analysis showed a relationship between 25(OH)D level and breast cancer only in the group of premenopausal women [31], while a recently completed randomized clinical trial did not show an effect of vitamin D supplementation (2000 IU per day for five years) on the incidence of invasive breast cancer [32]. Moreover, unfavorable results were also presented, showing a correlation between a high 25(OH)D level and increased risk of breast cancer in the European population [33], and a relationship between a high 25(OH)D level and worse survival prognosis of patients with breast cancer [34].

Currently, vitamin D supplementation is widely recommended. Thus, in light of the inaccurate results of clinical trials (favorable/no effect/unfavorable) and our previous studies (acceleration of metastasis in the 4T1 model by calcitriol), we decided to investigate the effect of calcitriol on the macrophage phenotype, as macrophages are considered the main fraction of immune cells to infiltrate the tumor [35]. Our study is the first to report the role of calcitriol in modifying the macrophage phenotype in an ex vivo model of mouse bone marrow-derived macrophages (BMDMs). Additionally, the use of conditioned media (CM) from 4T1 (tumorigenic, metastatic, excessive immune activation), 67NR (tumorigenic, non-metastatic), and Eph4-Ev (normal epithelial) cell cultures allowed analyzing the influence of cancer cells with different potential for activation of the immune system, tumorigenicity, and metastasis on the macrophage phenotype, in the presence and absence of calcitriol. Our results partially explain the phenomenon of calcitriol-stimulated metastasis which was observed in our previous studies in the in vivo 4T1 model and may indicate that the immunosuppressive properties of vitamin D may adversely affect the macrophages present in the tumor microenvironment.

## 2. Results

### 2.1. Calcitriol Does Not Significantly Affect the Proliferation of BMDMs

Differentiation of BMDMs was monitored under a light microscope (Appendix A). In the following days, we observed that the bone marrow progenitor cells gradually lengthened and differentiated, achieving a macrophage-like appearance around five–six days. After seven days, mature BMDMs were polarized to individual classes, with or without calcitriol. A tendency of enhanced proliferation was observed for BMDMs polarized in the presence of calcitriol; however, the differences were not statistically significant (Appendix A). Polarization of macrophages into appropriate classes led to noticeable morphological changes (Appendix A). Compared to the smallest, unpolarized M0 BMDMs, M1 and M2 cells were larger and more differentiated.

### 2.2. Calcitriol Differentially Alters the Expression of BMDM Genes

Using real-time PCR, mRNA expression was analyzed for the genes characteristic of M1 (*Cd80*—CD80 Molecule) and M2 (*Cd206*—Mannose Receptor C-Type 1) macrophages as well as for other examined genes (*Cox2*—Cyclooxygenase 2, *Spp1*—Osteopontin, *Vdr*—Vitamin D Receptor).

As expected, M1 BMDMs showed significantly higher expression of *Cd80* (Figure 1) compared to M0 and M2 BMDMs (*p* < 0.05; 2.1- and 3.0-fold, respectively), while M2 BMDMs had a higher level of *Cd206* mRNA compared to M0 and M1 cells (*p* < 0.05; 2.9- and 145.0-fold, respectively). Calcitriol reduced *Cd80* expression in M1 BMDMs (*p* < 0.05), while in the case of M2 BMDMs, a nearly 2.0-fold increase in *Cd80* expression was seen after stimulation with calcitriol. Furthermore, calcitriol increased the level of *Cd206* mRNA in both M0 and M2 BMDMs (*p* < 0.05).

M2 BMDMs showed almost two times higher expression of *Cox2* mRNA (Figure 1) than M0 and M1 cells (*p* < 0.05); however, stimulation with calcitriol did not significantly affect the expression of this gene in any class of BMDMs. In addition, M0 and M2 BMDMs had a similar level of expression of the *Spp1* gene, which was about 3.5 times higher than M1 macrophages (*p* < 0.05). Calcitriol significantly increased the mRNA expression of this gene in M0 and M2 macrophages (*p* < 0.05), whereas in M1 BMDMs, the opposite effect was observed (*p* < 0.05). Regarding *Vdr* expression, both M1 and M2 macrophages had a higher level of *Vdr* mRNA relative to the M0 class (*p* < 0.05; 13.8- and 19.2-fold, respectively). Calcitriol significantly increased *Vdr* expression by 2.5-fold in M0, 3.1-fold in M1, and 21.1-fold in M2 cells (*p* < 0.05).

### 2.3. Calcitriol Slightly Influences the Expression of BMDMs Proteins

Using fluorescence-activated cell sorting (FACS), we analyzed the expression level of surface markers that are characteristic of all macrophages (Pan-macrophage: CD11b—Integrin Alpha M; F4/80—EGF-like Module-Containing Mucin-like Hormone Receptor-Like 1; CD44), M1 (MHC II; CD54—Intercellular Adhesion Molecule 1; CD80—B7.1 Protein; CD86—B7.2 Protein), and M2 (CD163—Hemoglobin Scavenger Receptor; CD204—Macrophage Scavenger Receptor 1; CD36—Scavenger Receptor Class B Member 3). The gating strategy and images of representative histograms for each BMDMs class are provided in Appendix A.

More than 95% of BMDMs of all classes showed the expression of CD11b and F4/80 (Figure 2A), while a significantly higher percentage of CD44+ cells was found among M1 and M2 BMDMs (99.5%) compared to M0 BMDMs (*p* < 0.05; 96.2%). No significant differences were observed in mean fluorescence intensity (MFI) for F4/80, while M1 and M2 had significantly higher MFI for CD11b and CD44 (*p* < 0.05; 1.5- and 2.5-fold, respectively). As expected, significantly more MHC II+, CD80+, and CD86+ cells were recorded among M1 than among M0 and M2 BMDMs (Figure 2B; *p* < 0.05). In addition, M1 BMDMs had a higher MFI for MHC II, CD54, and CD86 than the M0 and M2 classes (*p* < 0.05). On the other hand, more M2 cells showed the expression of the CD163 marker (Figure 2C) and had a significantly higher MFI for CD204 and CD36 compared to M0 and M1 BMDMs (*p* < 0.05).

In the case of M1 macrophages, stimulation with calcitriol increased the expression of CD11b and MHC II (*p* < 0.05; 1.2- and 1.1-fold, respectively). However, we noticed a decrease in the percentage of the CD80+ population (*p* < 0.05; by 1.1%) and an increase in the percentage of the CD36+ population (*p* < 0.05; by 3.3%) in relation to M2 BMDMs.

Moreover, Western blot analysis (Appendix A; the uncropped Western Blots for individual proteins are available as Appendix A) showed that VDR was expressed at the protein level only in calcitriol-stimulated macrophages, especially in M2+cal macrophages (*p* < 0.05; 36.4-fold compared to M1+cal). CYP27B1 (Cytochrome P450 Family 27 Subfamily B Member 1) expression for M0 and M2 macrophages was at a similar level, while for M1 macrophages, it was 3.9-fold lower compared to M0 macrophages (*p* < 0.05). Calcitriol increased CYP27B1 expression only for M2 macrophages (2.4-fold; *p* < 0.05). The expression of CYP24A1 (Cytochrome P450 Family 24 Subfamily A Member 1) was approximately 2-fold lower for M1 and M2 macrophages compared to unstimulated M0 (*p* < 0.05); calcitriol did not affect the expression of this protein.

### 2.4. Calcitriol Modifies the Pattern of Cytokine/Chemokine Production Characteristic of Specific BMDMs Classes

A cytokine array allows detecting 40 different cytokines and chemokines. In this study, we noted the expression of 18 cytokines/chemokines (Figure 3) in at least one class of BMDMs.

In comparison with M0 BMDMs, both M1 and M2 BMDMs produced significantly more IL-1RA, CXL10, CXCL9, and TNF-α, but significantly less CXCL2 (*p* < 0.05). In addition, a significantly lower concentration of CXCL11, CXCL1, CCL3, and CCL4 was observed in the supernatants obtained from the M1 BMDMs culture compared to M0 BMDMs (*p* < 0.05). M2 macrophages were characterized by higher expression of GM-CSF, G-CSF, IL-10, IL-27, CXCL1, CCL12, and TIMP-1 relative to both M0 and M1 macrophages, as well as higher expression of CCL3, CCL4, and CXCL2 relative to M1 macrophages (*p* < 0.05). Furthermore, the supernatants obtained from the M2 BMDMs culture had significantly lower levels of IL-1RA, CXCL9, CXCL12, and TNF-α than the supernatants from the M1 BMDMs culture (*p* < 0.05).

Calcitriol significantly reduced the level of CXCL10 in M0 macrophages (*p* < 0.05; 1.6-fold) and the level of CCL3 in M2 macrophages (*p* < 0.05; 1.3-fold). M2 BMDMs stimulated with calcitriol secreted 1.2 times more CXCL1 than untreated cells (*p* < 0.05). In addition, calcitriol had a positive effect on the production of IFN-γ, IL-1RA, and CXCL11 in M1 BMDMs (increase by 50.0-, 1.3-, and 2.2-fold, respectively).

### 2.5. Calcitriol Reduces or Increases the Expression of Markers Characteristic of Class M1 or M2 BMDMs

As expected, the presence of nitrite ions, reflecting the level of nitric oxide (NO), was found only in the supernatants of M1 BMDMs (Figure 4A). On the other hand, the arginase activity characteristic of M2 BMDMs was significantly higher in M2 cells (*p* < 0.05; 4.2-fold) in relation to M1, while no activity was observed in unstimulated M0 macrophages (Figure 4B). Furthermore, the presence of IL-1 was not recorded in the supernatants of M0 BMDMs and that of IL-6 in the supernatants of M0 and M2 BMDMs (Figure 4C). M1 macrophages produced significantly more IL-6 and CCL2 than M2 cells (*p* < 0.05; 1.4- and 2.6-fold, respectively). The level of CCL2 was also 3.8-fold higher (*p* < 0.05) in M2 BMDMs cultures compared to M0 BMDMs. Additionally, M1 cells secreted significantly lesser osteopontin (OPN) compared to M0 and M2 cells (*p* < 0.05; 2.3- and 2.4-fold, respectively).

Interestingly, calcitriol significantly reduced the concentration of nitrite ions (Figure 4A) in M1 cultures (*p* < 0.05; 1.1-fold), but increased the arginase activity (Figure 4B) in both M1 and M2 cultures (*p* < 0.05; 1.9- and 1.5-fold, respectively). A significant decrease in the concentration of all tested cytokines (Figure 4C) was also noted in calcitriol-stimulated M1 BMDMs cultures (*p* < 0.05). In addition, calcitriol reduced the concentration of IL-6 in the supernatants of M2 BMDMs (*p* < 0.05; 1.1-fold), while an increase in CCL2 production was noted in M0 and M2 BMDMs (*p* < 0.05; 1.7-fold in both).

### 2.6. CM from Cancer and Normal Cells Affects the Proliferation of BMDMs Differently

CM of 4T1 cells, both alone and in combination with calcitriol, increased the proliferation of all classes of BMDMs compared to untreated BMDMs (*p* < 0.05; Figure 5). Subsequently, 4T1 CM alone increased the proliferation of M0 BMDMs by 29.5%, M1 BMDMs by 9.3%, and M2 BMDMs by 21.7% in relation to untreated M0. Addition of calcitriol to 4T1 CM significantly increased (*p* < 0.05) the proliferation of only M0 macrophages by 53.6% in comparison to M0 BMDMs incubated with 4T1 CM and by 83.1% in comparison to untreated M0.

CM of 67NR cells, both alone and in combination with calcitriol, increased the proliferation of M0 (by 28.5% and 36.1%, respectively) and M2 BMDMs (by 9.7% and 16.5%, respectively), but inhibited the proliferation of M1 BMDMs (by 9.1% and 3.7%), compared to untreated BMDMs (*p* < 0.05; Figure 5).

CM of Eph4-Ev cells affected the proliferation of only M2 macrophages, stimulating their growth by 10.9% (CM) and 18.6% (CM + calcitriol) in relation to untreated M2 macrophages (*p* < 0.05; Figure 5). Calcitriol did not additionally reduce or enhance the proliferation of BMDMs of any classes when used in combination with 67NR or Eph4-Ev CM.

### 2.7. Calcitriol Modulates CM-Induced Gene Expression of BMDMs

4T1 and 67NR CM significantly (*p* < 0.05; Figure 6) stimulated the mRNA expression of *Cd80* in M0 (1.7- and 1.6-fold, respectively) and M2 (2.8- and 2.0-fold, respectively) BMDMs. Conversely, M1 BMDMs incubated with each CM showed significantly lower expression of *Cd80* mRNA (2.1-, 1.6-, and 1.8-fold for 4T1, 67NR, and Eph4-Ev, respectively). Calcitriol in combination with 4T1 CM increased the expression of the *Cd80* gene in all BMDMs classes (*p* < 0.05, compared to CM stimulation without calcitriol). Furthermore, the addition of calcitriol to 67NR or Eph4-Ev CM caused an increase in the mRNA expression of *Cd80* in M1 (only Eph4-Ev CM) and M2 (both CM) macrophages and a decrease in expression in M0 (both CM) macrophages (*p* < 0.05).

In general, all CM significantly increased the expression of *Cd206* in all classes of BMDMs (*p* < 0.05; in the highest degree in M2 BMDMs: 4T1 CM 8.9-fold, 67NR CM 2.2-fold, and Eph4-Ev 3.8-fold), except for a significant (1.4-fold) reduction in M1 macrophages by Eph4-Ev CM (Figure 6). Calcitriol further upregulated the expression of this gene in combination with 4T1 CM in M0 and M1 BMDMs and with 67NR CM in M0 BMDMs (*p* < 0.05, compared to CM stimulation without calcitriol). Conversely, a decrease in expression was noted for the combination of calcitriol with 4T1 and Eph4-Ev CM in M2 macrophages and for the combination of calcitriol with 67NR in M1 macrophages (*p* < 0.05).

Only 4T1 CM significantly increased (*p* < 0.05) *Cox2* mRNA expression in M0 and M1 BMDMs (1.8-fold in both; Figure 6); however, the addition of calcitriol significantly lowered the expression of this gene (compared to CM stimulation without calcitriol).

The expression of *Spp1* mRNA was upregulated by each CM in M0 BMDMs (*p* < 0.05; average 2.4-fold; Figure 6), whereas in M1 BMDMs, a significant decrease in expression was observed (2.2-, 1.2-, and 1.7-fold for 4T1, 67NR, and Eph4-Ev, respectively). Furthermore, 4T1 and 67NR CM increased *Cox2* gene expression (3.4- and 1.9-fold, respectively) in M2 BMDMs (*p* < 0.05). The expression of this gene was significantly increased (*p* < 0.05) by the combination of calcitriol with 4T1 CM and Eph4-Ev CM in M1 and M2 BMDMs and by the combination of calcitriol with 67NR CM in M2 BMDMs, whereas downregulation was caused by calcitriol combined with Eph4-Ev CM in M0 BMDMs (all comparisons to CM stimulation without calcitriol).

All CM stimulated the level of *Vdr* mRNA in M0 BMDMs (*p* < 0.05; the highest degree by 4T1 CM—5.0-fold); by contrast, in M1 and M2 BMDMs, the expression of this gene was significantly decreased after stimulation with CM (on average 3.4- and 1.7-fold in M1 and M2 BMDMs, respectively; Figure 6). Calcitriol further increased the level of *Vdr* in combination with 4T1 CM in all macrophage classes, with 67NR in M0 and M2 BMDMs, and with Eph4-Ev in M1 and M2 BMDMs (all comparisons to CM stimulation without calcitriol).

### 2.8. Calcitriol in Combination with CM Modifies the Profile of Cytokines Secreted by BMDMs

As mentioned above (Figure 4A), nitrite ions were found only in the supernatants of M1 BMDMs. 4T1 and 67NR CM reduced the ion level by 1.2-fold (*p* < 0.05; Figure 7); the addition of calcitriol caused a further reduction in the level only in combination with 4T1 CM (*p* < 0.05, compared to CM stimulation without calcitriol).

In addition, IL-1 was detectable only in M1 BMDMs (Figure 4C). All CM significantly lowered (*p* < 0.05) the concentration of this cytokine, and its level was undetectable when 4T1 CM was used in combination with calcitriol and 67NR CM was used with or without calcitriol (Figure 7).

Although IL-6 was detected in M1 and M2 BMDMs (Figure 4C), stimulation of M2 macrophages with each CM reduced the expression of this cytokine to undetectable levels (data not shown). Similar to IL-1, all CM significantly lowered (*p* < 0.05) the level of IL-6 in M1 BMDMs to a nonmeasurable concentration, especially 67NR used with or without calcitriol and 4T1/Eph4-Ev CM used in combination with calcitriol (Figure 7).

In M0 BMDMs, a significant increase (*p* < 0.05) in the CCL2 concentration was observed after incubation with all CM (the highest degree with 4T1 CM—2.4-fold; Figure 7); the addition of calcitriol upregulated the expression of CCL2 in each combination in this class of macrophages (compared to CM stimulation without calcitriol). In M1 and M2 BMDMs, the production of this cytokine was stimulated after treatment with 4T1 and 67NR CM (the highest degree with 4T1 CM—1.7- and 2.2-fold in M1 and M2 BMDMs, respectively), and a decrease was observed after treatment with Eph4-Ev CM (1.5- and 2.4-fold in M1 and M2 BMDMs, respectively). Except with 4T1 CM in M1 macrophages, calcitriol significantly increased CCL2 levels in combination with each CM in all BMDMs classes (all comparisons to CM stimulation without calcitriol).

67NR reduced the level of OPN by 1.3-fold in M0 BMDMs (*p* < 0.05; Figure 7), while all CM significantly decreased the concentration of this protein in M2 BMDMs (the highest degree with Eph4-Ev CM—1.9-fold). By contrast, M1 BMDMs incubated with each CM secreted more OPN into the supernatant (*p* < 0.05). In M0 and M1 BMDMs, a decrease in the concentration of this protein was noted after treatment with calcitriol combined with 4T1 CM (M0), 67NR CM (M1), and Eph4-Ev CM (both M0 and M1), while M2 BMDMs showed higher OPN expression after treatment with calcitriol in combination with 4T1 CM (all comparisons to CM stimulation without calcitriol).

### 2.9. M2 BMDMs Differentiated in the Presence of Calcitriol Enhance the Migratory Potential of 4T1 and 67NR Cells

Due to the observed enhancement of M2 BMDMs polarization after treatment with calcitriol, we decided to investigate whether BMDMs of different classes (M0, M1, M2), polarized with the presence of calcitriol or without, could modify the migration properties of normal and cancer cells with different metastatic potential. To reflect on the interaction of macrophages with tumor cells, we generated CM of BMDMs, which we then used to stimulate the 4T1, 67NR, and Eph4-Ev cells. Two proteins that are components of the extracellular matrix—fibronectin and collagen IV—were used in the migration assay.

In the case of migration through collagen IV (Figure 8A), M2 CM increased the migration of 67NR cells 3.2 times compared to control cells (not treated with any BMDMs CM); M2+cal CM further enhanced the migration properties of 67NR cells (5.5-fold compared to control cells and 1.7-fold compared to cells incubated with M2 CM). Incubation of 4T1 cells with any of the BMDMs CM did not significantly affect cell migration through collagen IV (Figure 8A). However, analyzing migration through the fibronectin layer (Figure 8B), we noted that 4T1 cells migrated more readily after incubation with M2+cal CM (2.0-fold compared to control cells); *p* < 0.05). We found no significant differences in the migration of 67NR cells by fibronectin after incubation with BMDMs CM (Figure 8B). Interestingly, BMDMs CM, regardless of macrophage subtype, inhibited the migration of normal epithelial Eph4-Ev cells through both collagen IV and fibronectin (*p* < 0.05; Figure 8A,B). Calcitriol had no significant effect on the modification of the impact of BMDMs on the migratory properties of Eph4-Ev cells.

## 3. Discussion

IL-4/IL-13-induced M2 macrophages are the most commonly used in vitro and ex vivo models to compare the functions and properties of M2 macrophages with the classical M1 macrophages. In our study, IL-4 successfully induced M2 macrophages, while M1 macrophages were differentiated by the combination of LPS and IFN-γ (Figure 9). According to scientific reports, at the transcriptional level, M1 macrophages expressed a higher level of *Cd80* and a lower level of *Cd206*, while the opposite tendency was seen for M2 macrophages (Figure 1). Further, at the protein level, M1 macrophages showed higher expression of MHC II, CD54, and CD86, while M2 macrophages had higher levels of CD163, CD204, and CD36 (Figure 2). Overall, M2 macrophages secreted higher amounts of cytokines such as G-CSF, GM-CSF, IL-10, and IL-27, and chemokines such as CXCL1, CCL12, CCL3, CCL4, CXCL2, and TIMP-1, while M1 macrophages produced more IL-1, IL-1RA, IL-6, and TNF-α, as well as more CCL2, CXCL11, and CXCL9 (Figure 3 and Figure 4). Interestingly, CXCL1 and CXCL2, including those produced by TAMs, were identified as capable of contributing to metastasis and chemoresistance in a mouse spontaneous breast cancer model [36,37]. In addition, CCL2, CCL3, and CCL4 were reported as chemokines associated with worse prognosis and faster tumor progression [38,39]. On the other hand, CXCL9 and CXCL11, secreted by M1 macrophages, promote a Th1-type response and stimulate tumor-infiltrating cytotoxic T cells [40]. Moreover, in line with the results of other studies [4,9,41,42], in our experiments, M1 macrophages secreted nitrite ions into the supernatants, while M2 macrophages showed higher ARG1 expression (Figure 4).

For both M0 and M2 BMDMs, calcitriol increased the expression of the marker typical for alternative macrophages, *Cd206*, and decreased the potential to differentiate into M1 macrophages as shown by the measured *Cd80* expression (Figure 1). This is consistent with results of other studies carried out in other monocyte/macrophage models, in which calcitriol decreased the expression of CD80 [13] and increased CD206 [18,43]. At the protein level, we observed a decrease in the percentage of CD80+ cells and an increase in CD36+ cells in the M2 BMDMs, which also indicates an enhancement of M2 polarization. On the other hand, we also unexpectedly found an increase in MHC II expression in M1 BMDMs, which is contrary to the results of Xu et al. [44], where a decreased expression of this molecule on monocytes was noted after vitamin D treatment. However, in our experiments, these changes were related to expression intensity, while the percentage of MHC II+ M1 BMDMs remained unchanged. Moreover, calcitriol enhanced arginase expression in both M1 and M2 BMDMs and significantly reduced iNOS expression as measured by nitrite level in M1 BMDMs (Figure 4). Similarly, Chang et al. noted a reduction in the expression of iNOS mRNA and NO release in RAW 264.7 murine macrophages after treatment with LPS and calcitriol [15], whereas Scott et al. noted a significant increase in arginase expression in human CD163+ macrophages isolated from the skin of patients who received high levels of vitamin D_3_ after exposure to UV radiation at an erythema-inducing dose [19]. The properties and phenotypes of murine macrophages are related to the metabolism of arginine. Classical M1 macrophages express iNOS that converts arginine to citrulline and NO, which triggers antibacterial properties in this class of macrophages due to the activity of nitrogen species. On the other hand, M2 alternative macrophages convert arginine to ornithine and urea and further to polyamine and proline by arginase activity, stimulating the processes of proliferation and repair [4,45]. Moreover, in our study, calcitriol lowered the pro-inflammatory secretory properties of macrophages M1 and M2, as indicated by the production of IL-1 and IL-6, respectively (Figure 4). Interestingly, calcitriol slightly decreased the production of CCL2 by M1 BMDMs, while M2 macrophages produced more CCL2 after calcitriol treatment. CCL2 acts chemotactically on monocytes and macrophages, recruiting them to the site of inflammation. Xu et al. showed that CCL2 induces an M2-like phenotype in macrophages recruited to bone marrow in multiple myeloma [46]; similarly, Sierra-Filardi et al. showed that CCL2 targets the macrophages toward the M2 phenotype [47]. On the other hand, Carson et al. reported that CCL2 had no effect on the polarization of the phenotype to classical or alternative macrophages in the BMDMs model [48]. In the TME (tumor microenvironment), CCL2 can be produced by both cancer and stromal cells including TAMs and shows opposing pro- or antitumoral effects depending on its type and stage; however, high CCL2 expression is generally an unfavorable prognostic factor in different types of cancers [38,39,49].

It has been reported that CCL2, through cooperation with OPN, can promote lung metastasis in various mouse models [50]. Additionally, some authors have described the influence of OPN on the polarization of macrophages to an M2-like phenotype [51,52]. OPN is a molecule exhibiting pleiotropic effects, including both pro-inflammatory and anti-inflammatory, and contributing to cancer progression in different tumor models. In our previous studies, we showed that calcitriol did not affect the in vitro production of OPN by murine RAW 264.7 macrophages [30]. Interestingly, calcitriol increased the expression of *Spp1* mRNA in both M0 and M2 BMDMs, but not in M1 BMDMs, in which a reduction in the expression was noted (Figure 1). This effect was observed only at the protein level in the M1 class (Figure 4); however, OPN may also exist intracellularly in the cytoplasm and nucleus [53], so secretory OPN does not necessarily reflect OPN expression at the mRNA level. Thus, the results presented here indicate that in the model of mouse BMDMs, calcitriol enhances the polarization of the M2 phenotype, which was manifested by the increased expression of *Cd206* and *Spp1* mRNA and CCL2 in both M0 and M2 BMDMs and CD36 and Arg in M2 BMDMs and by decreased expression of *Cd80* and *Spp1* mRNA and IL-1, IL-6, OPN, and iNOS (inducible nitric oxide synthase) in M1 BMDMs.

In our previous studies, we indirectly showed that calcitriol may contribute to increased cancer progression in the 4T1 murine mammary gland cancer model by modifying the TME [24,28,29,54]. The results presented here show that calcitriol can support the ex vivo polarization of murine BMDMs to the M2 phenotype, which could explain the previously observed in vivo increase in the metastatic spread of 4T1 cells and 67NR following calcitriol treatment. Thus, in order to partially mimic the TME conditions in in vitro conditions, we generated CM from 4T1 metastatic cells, nonmetastatic 67NR cells, and normal epithelial Eph4-Ev cells and examined how CM alone and in combination with calcitriol affected some properties of BMDMs. Apart from the differences in invasiveness and metastasis, 4T1 tumors are characterized by high epithelial–mesenchymal plasticity [55] and engage and activate the host’s immune system, which is manifested by splenomegaly and leukocytosis, while in the 67NR model, such effects are not observed [56,57]. In our study, we found that 4T1 CM had the highest potential to stimulate the proliferation of macrophages, especially M0 and M2 BMDMs, and to a lesser extent M1 BMDMs (Figure 5). M2 BMDMs were stimulated independently by all CM including normal epithelial Eph4-Ev CM; however, the greatest stimulation was caused by 4T1 CM, while M1 BMDMs were stimulated only by this CM. This may indicate that in the TME, 4T1 cells can affect infiltrating TAMs regardless of their phenotype, thus altering their properties and functions. Moreover, we observed that calcitriol in combination with 4T1 CM enhanced the viability of unstimulated M0 macrophages, while a similar trend was also seen in M2 BMDMs for all CM used with calcitriol, which may indicate the higher immunomodulatory potential of calcitriol in the crosstalk between cancer cells and M2 macrophages than between cancer cells and M1 macrophages in the TME. In the context of the expression of markers characteristic of M1/M2 macrophages, we also observed the greatest differences after stimulation with 4T1 CM (Figure 6). Namely, 4T1 CM reduced *Cd80* expression in M1 BMDMs and strongly increased *Cd206* expression in M2 BMDMs; however, calcitriol reversed this trend to some extent. In relation to unstimulated M0 BMDMs, calcitriol in combination with 4T1 CM and, to a lesser extent, with 67NR CM increased *Cd206* expression which may suggest targeting their expression profile more towards M2 than M1. Interestingly, normal cells also upregulated *Cd206* expression but only in class M2. Additionally, we observed the synergistic effect of calcitriol and all CM (the highest grade for 4T1 CM) on the increase of *Spp1* expression in M2 macrophages, whereas at the protein level, this trend was (to a limited extent) only visible for 4T1 CM (Figure 7), which may be explained by the presence of an intracellular form of OPN. Madera et al. showed that unstimulated BMDMs exposed to 4T1 CM released higher amounts of pro-inflammatory cytokines such as TNF-α, IL-6, CCL2, and NO in response to LPS [58]. However, in our study, nitrite ions and IL-6 were undetectable in unstimulated M0 BMDMs (Figure 7). This difference may have been caused due to the fact that we did not use LPS as a stimulant for all macrophage classes in our study, so the cytokine levels that we observed are more physiological. We noticed that 4T1 CM decreased NO production in M1 BMDMs, and calcitriol further exacerbated this effect and thus decreased the M1 polarization potential. In addition, 4T1 and 67NR CM reduced the secretion of IL-1 and IL-6 by M1 macrophages and IL-6 secretion by M2 macrophages. Interestingly, 4T1 CM, to the greatest extent, increased the secretion of CCL2 by all macrophage classes, which in the TME may cause the activation and acceleration of the CCL2–CCR2 axis to stimulate the metastatic process. Additionally, calcitriol further increased the potential of M0 and M2 BMDMs to secrete CCL2 in combination with 4T1 CM, while the opposite effect was seen for M1 BMDMs. This phenomenon may also indicate that calcitriol is effective for enhancement of M2 polarization, especially in the 4T1 model. Thus, in our study, we found that 4T1 CM (metastatic) had the greater potential to affect gene and protein expression in BMDMs than 67NR CM (nonmetastatic) and Eph4-Ev CM (normal), with the greatest effect seen in M2 macrophages which increased their differentiation and properties characteristic of alternative macrophages.

Moreover, we also showed that M2 BMDMs differentiated in the presence of calcitriol stimulated migration of 4T1 cells through fibronectin and migration of 67NR through collagen type IV (Figure 8), while migratory properties of normal epithelial Eph4-Ev cells were reduced after incubation with BMDMs CM regardless of macrophage class and the presence of calcitriol in the differentiation medium. It has been proven that M2 macrophages can promote migration in various tumor models [59,60,61], while calcitriol, by acting directly on cancer cells, may reduce their ability to migrate and invade [62,63]. In our research, we have shown that calcitriol, by enhancing the M2 phenotype of BMDMs, contributes to the intensification of the stimulating effect of M2 macrophages on the migration properties of cancer cells. Thus, it proves that calcitriol can affect tumor progression on many levels, including modifying the TME, which may explain the adverse effects of vitamin D treatment in animal studies observed by some authors. The phenomenon of the enhancement of the M2 phenotype by calcitriol could be explained by the differential expression of the vitamin D receptor in individual classes of macrophages, which could cause their different responses to calcitriol. VDR has been shown to be crucial for the antimicrobial properties of macrophages; however, its expression level in different macrophage classes has not been investigated so far [64]. In our study, we found that the expression of *Vdr* mRNA was the highest in M2 BMDMs (Figure 1). Moreover, although calcitriol increased the expression of *Vdr* in all macrophage classes, the greatest upregulation was again observed in M2 BMDMs. Interestingly, the combination of 4T1 CM with calcitriol dramatically increased *Vdr* expression in M0 BMDMs to the level observed in M2 macrophages, which was not observed with other CM combinations. On the other hand, at the protein level, VDR expression was found only in calcitriol-stimulated macrophages, which was again significantly higher in M2 BMDMs (Appendix A). Furthermore, calcitriol highly stimulated the protein expression of CYP27B1, without affecting the expression of CYP24A1—vitamin D-metabolizing enzymes (Appendix A): CYP27B1, 1α-hydroxylase, is responsible for calcitriol synthesis from its precursor 25(OH)D_3_; CYP24A1 (24-hydroxylase), on the other hand, degrades all vitamin D metabolites to inactive forms [65]. Thus, this phenomenon leads to an increase in the local synthesis of calcitriol in M2 macrophages, with highly increased expression of VDR, which may indicate that alternative macrophages may be more sensitive to the immunomodulatory effects of calcitriol. Therefore, it seems that calcitriol may intensify the immunosuppression of the tumor niche, which in specific cases may support the unfavorable phenotype of immune cells present in the TME, including macrophages, contributing to the stimulation of cancer progression. The observed phenomenon of enhancement of the alternative macrophage phenotype by calcitriol, also in the presence of 4T1 CM, may partially explain the discrepancies in clinical trials (favorable/no effect/unfavorable) concerning the effect of vitamin D supplementation on the risk and course of invasive breast cancer. However, further studies are needed to clarify whether the adverse effect of vitamin D observed in some preclinical and clinical studies is related only to cancers with excessive pathological activation of the immune system, at a specific stage of advancement or with a specific composition of the TME.

## 4. Materials and Methods

### 4.1. Ex Vivo Differentiation of BMDMs and Phenotype Polarization

To differentiate BMDMs, progenitor bone marrow cells were collected from 8- to 12-week-old C57BL/6/FoxP3^GFP^ male mice which were obtained from Experimental Animal Facility IIET (Wroclaw, Poland). The animals were anesthetized using a 3–5% (*v*/*v*) mixture of isoflurane (Aerrane Isofluranum, Baxter, Deerfield, MA, USA) and then euthanized by cervical dislocation by qualified personnel, in accordance with the EU Directive 2010/63/EU on the protection of animals used for scientific purposes. As we received the mice intended for euthanasia, in accordance with the “Directive 2010/63/EU of the European Parliament and of the Council, of 22 September 2010, on the protection of animals used for scientific purposes”, our experiments did not require the approval of the local ethics committee. After euthanasia, the skin around the hind legs was flushed with ethanol and removed along with muscle tissue using scissors and a scalpel. Femur and tibial bones of both legs were cut off from the body, keeping the hip joint intact, and were placed in cold DMEM (Thermo Fisher Scientific, Waltham, MA, USA) with antibiotics. After keeping on ice, femurs and tibias were transferred to a sterile petri dish, cleaned of remaining muscle tissue, and then cut on both sides. Each bone was flushed at both sides with 5 mL of cold phosphate-buffered saline (PBS; IIET, Wroclaw, Poland). The obtained cell suspension was filtered by a cell strainer with a mesh size of 70 µm (Greiner Bio-One, Kremsmunster, Austria).

The prepared bone marrow cell suspension was centrifuged (432× *g*, 7 min, 4 °C) and resuspended in DMEM containing 10% fetal bovine serum (FBS), 4.5 g/L glucose, 4.0 mM L-glutamine, 1 mM sodium pyruvate, 1× nonessential amino acids, 100 µg/mL streptomycin (all Sigma-Aldrich, Saint-Louis, MO, USA), and 100 U/mL penicillin (Polfa Tarchomin S.A., Warsaw, Poland). Cell quantity and viability were determined by counting in a Bürker chamber in a trypan blue solution (Sigma-Aldrich, Saint-Louis, MO, USA). Around 2 × 10^6^ viable bone marrow cells were resuspended in 4 mL/well of medium supplemented with 25 ng/mL of mouse recombinant M-CSF (BioLegend, San Diego, CA, USA) and 0.05 mM 2-mercaptoethanol (Sigma-Aldrich, Saint-Louis, MO, USA) (Day 0) on a six-well plate. BMDMs were cultured at 37 °C in a humidified atmosphere with 5% CO_2_. On Days 3 and 5, fresh medium was prepared with M-CSF and 2-mercaptoethanol and mixed in a 1:1 ratio with the medium collected from wells containing differentiating macrophages. On Day 7, the cultures formed monolayers and no nonadherent cells were observed. The purity of BMDMs was determined in a flow cytometer by examining the expression of the markers CD11b and F4/80 (Section 4.4).

To induce phenotype polarization, BMDMs were washed with PBS and transferred to a medium containing appropriate cytokines, with 50 ng/mL mouse recombinant IFN-γ (BioLegend, San Diego, CA, USA) and 100 ng/mL LPS (Sigma-Aldrich, Saint-Louis, MO, USA) for M1 BMDMs or 20 ng/mL mouse recombinant IL-4 for M2 BMDMs. The control was M0 BMDMs without additional stimulating factors. Calcitriol (Cayman Chemical, Ann Arbor, USA) was added simultaneously with appropriate cytokines at a concentration of 100 nM, and polarization was carried out for 48 h. Then, differentiated BMDMs were washed with PBS and used for further analysis. The isolation of bone marrow progenitor cells and their differentiation toward macrophages is shown in Figure 9.

### 4.2. SRB Cell Proliferation Assay

After differentiation, the effect of calcitriol stimulation on the proliferation of various classes of BMDMs was determined using the SRB assay as described previously [66]. Immediately after the SRB protocol, the absorbance was measured by a Synergy H4 plate reader (BioTek, Winooski, VT, USA) at a wavelength of 540 nm on a six-well plate. The effect of calcitriol on proliferation was compared only among BMDMs of individual classes—M0, M1, and M2—while cells that were not stimulated with calcitriol were treated as control (100%).

### 4.3. RNA Isolation and Real-Time PCR Analysis

BMDMs of individual classes, incubated with or without calcitriol, were washed with PBS and resuspended in TRI Reagent solution (Molecular Research Center, Cincinnati, OH, USA). RNA was isolated using dedicated columns (A&A Biotechnology, Gdynia, Poland) according to the manufacturer’s instructions. The purity of isolated RNA was checked by spectrophotometric analysis in a Nanodrop 2000 spectrophotometer (Thermo Fisher Scientific, Waltham, MA, USA) at a wavelength of 260 nm. RNA samples were purified from genomic DNA by incubating with DNase (Thermo Fisher Scientific, Waltham, MA, USA) in the presence of RNase inhibitors (EURx, Gdansk, Poland). Then, the purified RNA was transcribed into cDNA using reverse transcriptase (Bio-Rad, Hercules, CA, USA).

Gene expression was studied in a ViiA™ 7 Real-Time PCR System (Thermo Fisher Scientific, Waltham, MA, USA) using Taq-Man chemistry and probes specific for the following genes: *Cd80* (Mm00711660_m1), *Cd206* (Mm01329362_m1), *Spp1* (Mm00436767_m1), *Cox2* (Mm03294838_g1), and *Vdr* (Mm00437297). Briefly, for a single reaction, 50 ng cDNA was used, and each sample was prepared in three technical replicates. Each amplification cycle was performed at 95 °C for 15 s and at 60 °C for 1 min (total 40 cycles). The relative quantification level of the examined gene expression, referred to as fold change, was calculated based on changes in the ΔΔCt values of the studied genes in relation to the control housekeeping gene *Hprt1* (Mm00446968) using DataAssist 3.01 software (Thermo Fisher Scientific, Waltham, MA, USA). The endogenous control gene was selected from 32 potential housekeeping genes as the most stable among the samples tested (TaqMan™ Array Mouse Endogenous Controls Plate; Thermo Fisher Scientific, Waltham, MA, USA).

### 4.4. Cell Surface Marker Analysis by Flow Cytometry

BMDMs of individual classes, stimulated with or without calcitriol, were washed with PBS and incubated for 1 h with Accutase solution (Sigma-Aldrich, Saint-Louis, MO, USA) for nonenzymatic cell detachment. Then, the well contents were pipetted, washed with PBS, and the obtained cell suspension was transferred to fresh tubes. Cells were counted in a Bürker chamber in trypan blue to control cell viability. Before cytometric staining, 1 × 10^5^ BMDMs were centrifuged (324× *g*, 7 min, 4 °C) and resuspended in serum-free PBS solution containing Fixable Viability Dye eFluor 780 (0.1 µL/100 µL volume; Thermo Fisher Scientific, Waltham, MA, USA) to distinguish dead cells from live cells. Cells were incubated for 30 min, at 4 °C, in the darkness, and then centrifuged again in PBS containing 2% FBS. The obtained cells were suspended in 2% PBS containing TruStain FcX (anti-mouse CD16/CD32) antibody (BioLegend, San Diego, CA, USA) and incubated for 10 min at 4 °C to block the Fc receptors (0.1 µg/100 µL volume). The BMDMs were then centrifuged and resuspended in 100 µL of 2% PBS containing fluorochrome-conjugated antibodies or the appropriate isotype control at a concentration recommended by the manufacturer. Two separate antibody mixes were prepared (details shown in Table 2), and the cells were incubated for 30 min at 4 °C in the darkness. Then, the BMDMs were centrifuged again and resuspended in 200 µL of 2% PBS for analysis.

Compensations were carried out for both sets of antibodies. Data reading and analysis were performed using a BD LSR Fortessa cytometer with FACSDiva V8.0.1 software (BD Biosciences, Franklin Lakes, NJ, USA). The percentage of positive cells and the MFI of stained cells in relation to the isotype control were determined. Cytometric analysis was carried out for BMDMs generated from four mice (number of independent repetitions = 4).

### 4.5. Cytokine Array

BMDMs of individual classes, incubated with or without calcitriol, were washed twice with PBS solution and incubated for 24 h in a proper medium without FBS (cell starvation). Then, the medium was collected and centrifuged (12,000× *g*, 10 min, 4 °C) to remove cell debris. Supernatants were transferred to fresh tubes and used in the cytokine array, ELISA (Section 4.6), and Griess test (Section 4.7).

Commercial Mouse Cytokine Array Panel A kits (Bio-Techne, Minneapolis, MN, USA) were used to detect 40 different chemokines and cytokines (each in two technical replicates). For this assay, 1 mL of generated medium per membrane was used, and the protocol was carried out according to the manufacturer’s instructions. Signal detection based on chemiluminescence was performed using a ChemiDoc Imaging System (Bio-Rad, Hercules, CA, USA), and the signal was evaluated using Bruker MISE software. Analysis was carried out for BMDMs generated from four mice (number of independent repetitions = 4).

### 4.6. Quantitative Protein Evaluation by ELISA

The following proteins were quantified by ELISA: OPN (Bio-Techne, Minneapolis, MN, USA); and IL-1, IL-6, and CCL2 (eBioscience, Thermo Fisher Scientific, Waltham, MA, USA). The protocols were carried out in accordance with the manufacturer’s instructions. Absorbance was measured in a Synergy H4 plate reader, and data analysis was performed using the CurveExpert ver. 1.4 software. Analysis was carried out for BMDMs generated from three mice (number of independent repetitions = 3).

### 4.7. Spectroscopic Determination of Nitrite Ions in the Griess Test

iNOS activity was determined using the Griess diazotization reaction-based assay [67]. Briefly, the BMDM supernatants (50 µL) were transferred to a 96-well plate, and 100 µL of Griess reagent was added (Sigma-Aldrich, Saint-Louis, MO, USA). Samples were incubated for 15 min in the dark (room temperature), and then absorbance was measured at 570 nm. Nitrite concentration was calculated using the sodium nitrite (Sigma-Aldrich, Saint-Louis, MO, USA) standard curve. Analysis was carried out for BMDMs generated from four mice (number of independent repetitions = 4).

### 4.8. Total Protein Quantification and Western Blot Assay

BMDMs of individual classes, incubated with or without calcitriol, were washed with PBS and suspended in RIPA lysis buffer with protease and phosphatase inhibitors (all Sigma-Aldrich, Saint-Louis, MO, USA). After 25 min of incubation on ice, the lysates were harvested using cell scrapers and transferred to fresh tubes. The obtained samples were centrifuged (12,000× *g*, 10 min, 4 °C) to get rid of cell debris. Then, supernatants were transferred to new tubes, frozen in liquid nitrogen, and placed at −80 °C for further use.

Protein concentration was determined with the DC Protein assay (Bio-Rad, Hercules, CA, USA). For this, the samples were thawed, centrifuged, and transferred to fresh tubes, and protein concentration was determined in accordance with the manufacturer’s instructions. BMDM lysates were used for Western blot analysis and measuring arginase activity (Section 4.9).

Briefly, 40 µg of protein was mixed with 4× Laemmli sample buffer (Bio-Rad, Hercules, CA, USA). Sodium Dodecyl Sulfate (12%) PolyAcrylamide Gel Electrophoresis (reagents: Sigma-Aldrich, Saint-Louis, MO, USA) was carried out in accordance with the guidelines. The correctness of electrophoretic separation was controlled by using the PageRuler™ Prestained Protein Ladder (Thermo Fisher Scientific, Waltham, USA). Then, the samples were transferred from the gel to Immobilon^®^-FL PVDF membranes (0.45 μm; Sigma-Aldrich, Saint-Louis, MO, USA). The membranes were blocked for 1 h at room temperature in 5% nonfat dry milk in 0.1% Tris-Buffered Saline/Tween-20 (TBS-T; IIET, Wroclaw, Poland/Sigma-Aldrich, Saint-Louis, MO, USA). After blocking, the membranes were washed (4 × 5 min in 0.1% TBS-T) followed by overnight incubation at 4 °C with rabbit anti-VDR monoclonal antibody (1:1000, D2K6W; Cell-Signaling, Danvers, USA), rabbit anti-CYP24A1 polyclonal antibody (1:1000, ab203308; Abcam, Cambridge, UK), or rabbit anti-CYP27B1 monoclonal antibody (1:1000, ab206655; Abcam, Cambridge, UK). Then, the membranes were washed and incubated for 1 h at room temperature with the secondary mouse antirabbit immunoglobulin G–horseradish peroxidase (HRP) monoclonal antibody (1:10,000; Santa Cruz Biotechnology Inc., Dallas, TX, USA). After subsequent washing, chemiluminescence was triggered using Clarity Western ECL Substrate (Bio-Rad, Hercules, CA, USA), and detection was performed in a ChemiDoc Imaging System (Bio-Rad, Hercules, CA, USA). After detection, the membranes were incubated for 30 min with 100% methanol at room temperature (Avantor Performance Materials Poland, Gliwice, Poland) to remove bound antibodies. Then, the membranes were washed, blocked for 1 h, washed again, and incubated with mouse anti-β-actin-HRP (C4) monoclonal antibody (1:1000; Santa Cruz Biotechnology Inc., Dallas, TX, USA) for 1 h at room temperature. Detection was carried out as described above. Densitometry analysis was performed using ImageJ software, and the expression of the tested proteins was normalized to that of β-actin. Analysis was carried out for BMDMs generated from four mice (number of independent repetitions = 4).

### 4.9. Determination of Arginase Activity

Arginase activity was measured using the Arginase Activity Assay Kit (Sigma-Aldrich, Saint-Louis, MO, USA) based on the colorimetric method. Briefly, 5 µg of BMDM lysate was applied to two wells in a 96-well plate, one for the sample well and one for the sample blank well. The sample volume was equalized with deionized H_2_O, and wells with 1 mM urea standard working solution and deionized H_2_O were also prepared. A 5× substrate buffer consisting of arginine buffer and Mn solution was added to the sample wells, and the arginase reaction was carried out for 1 h at 37 °C. To stop the reaction, 200 µL of the prepared urea reagent was added to each well (urea standard, H_2_O, sample, and sample blank wells). Then, 5× substrate buffer was added to the sample blank wells, and after incubating for 5 min at room temperature, the absorbance was measured at 430 nm. Based on the absorbance values, arginase activity was calculated using the following Equation (1):(1)$$\mathrm{Activity}=\frac{\left(A430\right)\mathrm{sample}-\left(A430\right)\mathrm{blank}}{\left(A430\right)\mathrm{standard}-\left(A430\right)\mathrm{water}}\text{×}\frac{{(1\;\mathrm{mM}\text{×}50\text{×}10}^{3})}{\left(V\text{×}T\right)}$$
where *T* = reaction time in minutes (120 min); *V* = sample volume (40 µl); 1 mM = concentration of urea standard; 50 = reaction volume (µL); and 10^3^ = mM-to-µM conversion. Analysis was carried out for BMDMs generated from four mice (number of independent repetitions = 4).

### 4.10. Imaging by Immunofluorescence Microscopy

For imaging, 10^4^ cells/well were cultured on a Falcon^®^ 96-well Black/Clear Flat Bottom TC-treated Imaging Microplate (Corning, New York, NY, USA), following the procedure described above. On the day of staining, cells were washed twice with PBS solution, fixed for 10–15 min in freshly prepared 4% paraformaldehyde (Avantor Performance Materials Poland, Gliwice, Poland), and permeabilized in 0.25% Triton X-100 (Sigma-Aldrich, Saint-Louis, MO, USA) for 15 min at room temperature. After two washes in PBS solution, cells were blocked for 30 min in 1% bovine serum albumin (Sigma-Aldrich, Saint-Louis, MO, USA) solution in 0.1% PBS/Tween 20 at room temperature. Then, the cells were rinsed with PBS three times (5 min each) and counterstained with DAPI (1:1500; Cell-Signaling, Danvers, TX, USA) and DyLight™ 554 Phalloidin (1:100; Cell-Signaling, Danvers, MA, USA) in PBS solution for 15 min at room temperature. Cells were reviewed and photographed under an Olympus IX81 fluorescent microscope (Olympus, Warsaw, Poland) with CellSense software (Olympus, Warsaw, Poland).

### 4.11. Generation of CM from Murine Breast Cancer and Normal Epithelial Cell Cultures

To determine the effect of mouse mammary gland cancer cells with diverse metastasis potential (4T1 vs. 67NR; metastatic vs. nonmetastatic; in comparison to normal epithelial cells Eph4-Ev) on the phenotype polarization status of BMDMs, CM were generated from cancer and normal cultures. Cells were grown in appropriate medium on a six-well plate to reach confluence (96 h) (4T1: RPMI Medium 1640—GlutaMAX), supplemented with 10% FBS HyClone (both Thermo Fisher Scientific, Waltham, MA, USA), 3.5 g/L glucose, 1 mM sodium pyruvate, 100 µg/mL streptomycin (all Sigma-Aldrich, Saint-Louis, MO, USA), and 100 U/mL penicillin (Polfa Tarchomin S.A., Warsaw, Poland); 67NR: DMEM supplemented with 10% FBS, 2.0 mM L-glutamine, 1× nonessential amino acids, 100 µg/mL streptomycin (all Sigma-Aldrich, Saint-Louis, MO, USA), and 100 U/mL penicillin (Polfa Tarchomin S.A., Warsaw, Poland); Eph4: DMEM supplemented with 10% FBS, 4.0 mM L-glutamine, 1.2 mg/L puromycin dihydrochloride, 100 µg/mL streptomycin (all Sigma-Aldrich, Saint-Louis, MO, USA), and 100 U/mL penicillin (Polfa Tarchomin S.A., Warsaw, Poland). 4T1 and Eph4-Ev cells were obtained from American Type Culture Collection (ATCC, Manassas, USA), and the 67NR cells were procured from the Barbara Ann Karmanos Cancer Institute (Detroit, MI, USA).

After the monolayer was formed, the wells were washed with PBS, and the cell cultures were incubated with an appropriate medium without FBS for 24 h (for conditioning of media). Then, the medium was collected, centrifuged (12,000× *g*, 10 min, 4 °C), and applied to BMDM cultures during macrophage polarization (with appropriate cytokines, with or without calcitriol). Polarization was carried out for 48 h in the presence of 35% CM from tumor or normal cells, and the effects of CM on proliferation (Section 4.2), gene expression (Section 4.3), and cytokine/chemokine production (Section 4.6) were studied. Analysis was carried out for BMDMs generated from three mice (number of independent repetitions = 3).

### 4.12. The Transwell Migration Assay

BMDMs of individual classes, incubated with or without calcitriol (Figure 9), were washed twice with PBS solution and incubated for 24 h in DMEM without FBS (cell starvation). Then, the medium was collected and centrifuged (12,000× *g*, 10 min, 4 °C) to remove cell debris. Supernatants were transferred to fresh tubes and used as conditioned media (CM) of BMDMs (M0, M0+cal, M1, M1+cal, M2, M2+cal, respectively).

4T1 (4000 cells per well), 67NR (7000 cells per well), and Ep4-Ev cells (3000 cells per well) were seeded in a 24-well plate in 0.5 mL of the appropriate medium. After 24 h, 0.5 mL of the appropriate CM of BMDMs was added to the cells and incubated for another 72h. The control of the assay was cells incubated with 0.5 mL of fresh DMEM without serum.

Inserts (24-well plate format, 6.5 mm inserts, 8.0 pore size, VWR International, Radnor, PA, USA) were coated overnight at 4 °C with a fibronectin or collagen type IV solution (Sigma-Aldrich, Saint-Louis, MO, USA) at a concentration of 10 µg/mL, dissolved in water or 0.5 M acetic acid, respectively. The next day, the coated inserts were rinsed twice with PBS solution and blocked with 1% BSA solution (Sigma-Aldrich, Saint-Louis, MO, USA) for 1 h at 37 °C. After this time, the inserts were washed twice with PBS solution and placed in a sterile 24-well plate containing 750 µL per well of DMEM with 10% FBS (bottom). A total of 15,000 cells were suspended in 200 µL of serum-free DMEM and plated on the insert (upper chamber) and cell migration was performed in an incubator at a 37 °C for 8h. After incubation, the inserts were wiped inside with a cotton swab and rinsed with PBS solution, and cells that migrated to the other side of the insert were fixed and stained with the Diff-Quick-Set (Medion Diagnostics, Gräfelfing, Germany) according to the manufacturer’s instructions. Purple-stained cells were counted from the surface of the entire insert on the Olympus IX81 microscope (Olympus, Warsaw, Poland) at 40× magnification. The migration assay was repeated 4 times for each cell line.

### 4.13. Statistical Analysis

Statistical analysis was carried out using GraphPad Prism 7.01 (GraphPad Software Inc., San Diego, CA, USA). Shapiro–Wilk’s normality test and Bartlett’s test were performed to check the assumptions for analysis of variance. The tests used for the analysis of each set of data are indicated in the figure legends. A *p*-value of < 0.05 was considered significant.

## 5. Conclusions

Our results indicate that supplementation with vitamin D or adjuvant therapy with calcitriol derivatives in the course of neoplastic diseases may not always be beneficial, especially in the case of cancers causing excessive, pathological activation of the immune system. This may be due to the fact that vitamin D has immunosuppressive properties, which may lead to unfavorable stimulation of immune cells present in the TME, including macrophages, to immunosuppressive phenotypes supporting cancer progression.

## Figures and Tables

**Figure 1 cancers-12-03485-f001:**
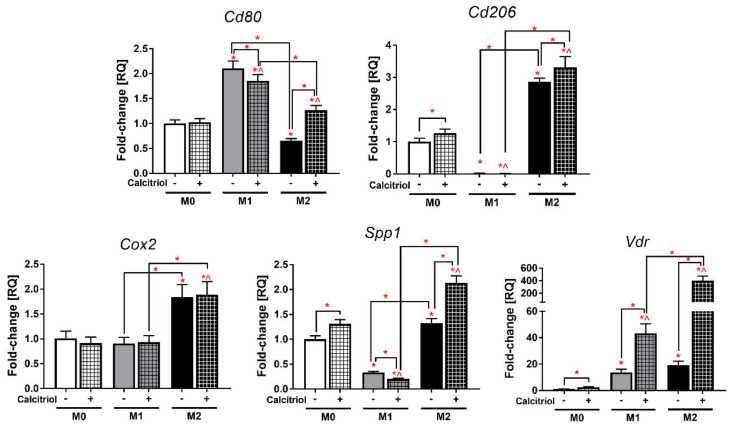
The effect of calcitriol on gene expression in individual classes of bone marrow-derived macrophages (BMDMs) analyzed by real-time PCR. Calcitriol lowers *Cd80* and *Spp1* expression for M1 BMDMs, increases *Cd80* expression for M2 BMDMs and *Cd206* and *Spp1* for M0 and M2 BMDMs, and increases *Vdr* expression in all BMDMs classes. The level of gene expression is presented in relation to the results obtained for M0 BMDMs (unstimulated and untreated with calcitriol). Briefly, RNA was isolated, purified, and transcribed into cDNA. Real-time PCR was performed using Taq-Man chemistry. A single reaction was performed with 50 ng of cDNA and each sample was performed in technical triplicates. The relative quantification (RQ) level of examined gene expression, referred to as fold change, was calculated based on the differences in ΔΔCt values of the studied genes in relation to the control housekeeping gene *Hprt1*. Data presentation: mean with standard deviation. Number of independent repetitions = 3 (BMDMs cultures generated from three mice). Statistical analysis: Sidak’s or Dunn’s multiple comparisons test. * *p* < 0.05 as compared to M0, ^ *p* < 0.05 as compared to M0 + cal or as indicated.

**Figure 2 cancers-12-03485-f002:**
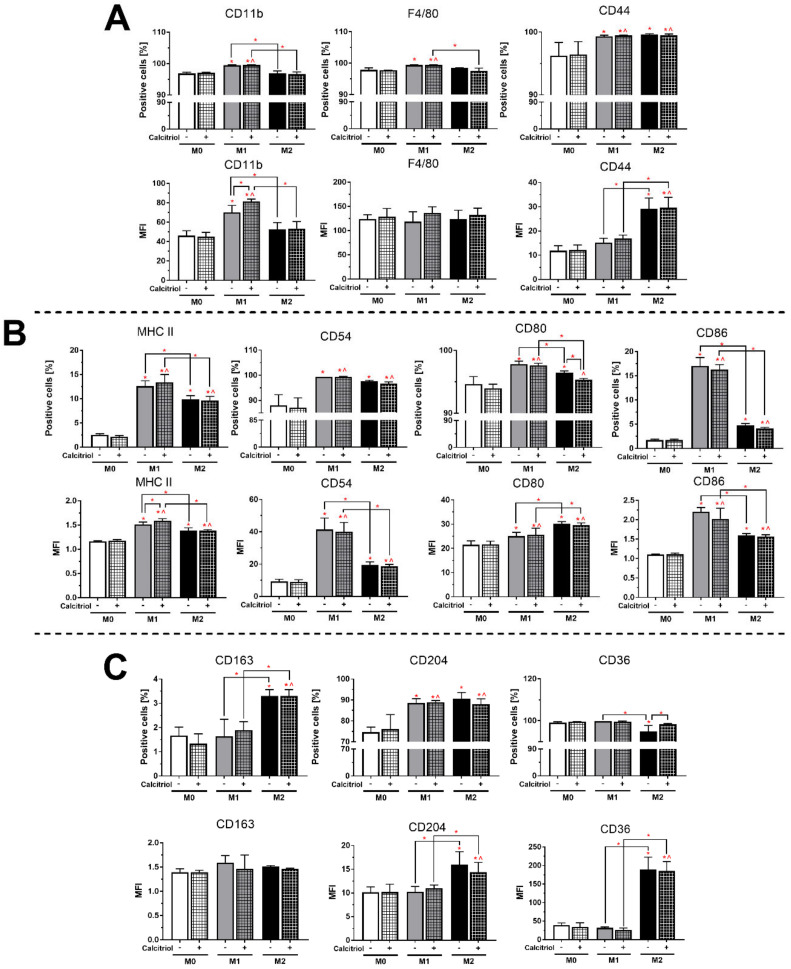
The effect of calcitriol on the expression of surface markers in individual classes of BMDMs analyzed by FACS: (**A**) Pan-macrophage markers, (**B**) M1 markers, and (**C**) M2 markers. Calcitriol enhances CD11b and MHC II expression for M1 BMDMs and lowers CD80 expression and increases CD36 expression for M2 BMDMs. Briefly, BMDMs were detached with a non-enzymatic solution. Then, 0.1 × 10^5^ cells were incubated with anti-mouse CD16/32 for blocking non-specific binding of immunoglobulin to the Fc receptors. Next, individual antibodies or isotype controls were applied and Fixable Viability Dye eFluor 780 was applied for viability control. The results are presented as percentage of positive cells expressing the examined molecule and as median fluorescence intensity (MFI). The gating strategy and images of representative histograms for each BMDMs class are provided in Appendix A. Data presentation: mean with standard deviation. Number of independent repetitions = 3 (BMDMs cultures generated from three mice). Statistical analysis: Sidak’s or Dunn’s multiple comparisons test. * *p* < 0.05 as compared to M0, ^ *p* < 0.05 as compared to M0 + cal or as indicated.

**Figure 3 cancers-12-03485-f003:**
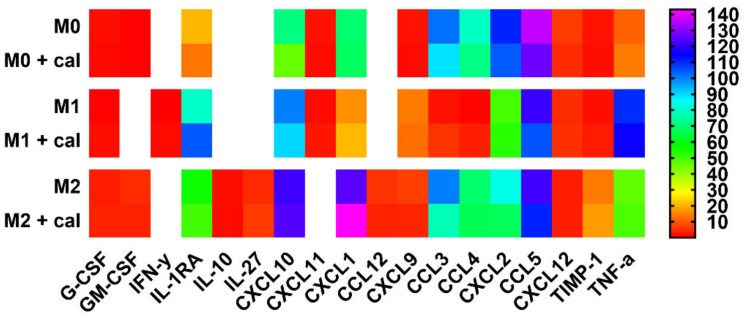
The influence of calcitriol on the production of cytokines and chemokines in the supernatants of BMDMs cultures analyzed by Mouse Cytokine Array Panel A assay. The macrophages M1 and M2 show a different profile of secreted chemokines and cytokines; calcitriol reduces the level of CXCL10 in M0 BMDMs and CCL3 in M2 BMDMs and stimulates the production of IFN-γ, IL-1RA, and CXCL11 in M1 BMDMs. Briefly, samples and membranes were incubated with the detection antibody cocktail overnight. The next day, the secondary antibody was applied and chemiluminescence was detected. The level of expression is presented as a heat map based on the results presented in Table 1. Data presentation: mean. Number of independent repetitions = 4 (BMDMs cultures generated from four mice).

**Figure 4 cancers-12-03485-f004:**
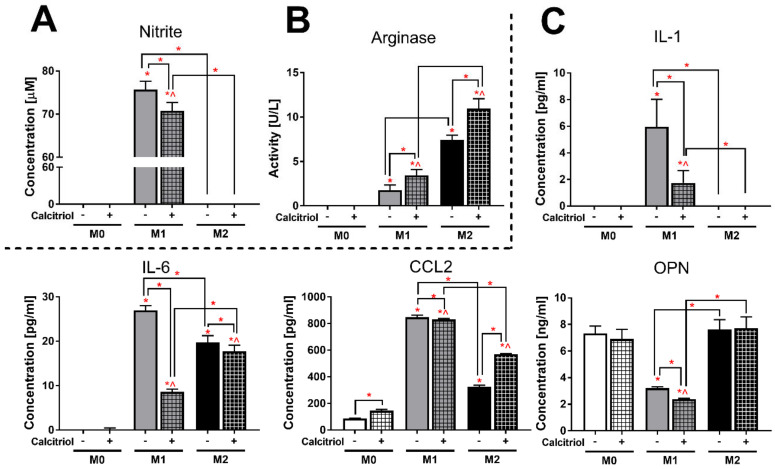
Determination of the effect of calcitriol on the level of expression of markers describing individual BMDMs classes: (**A**) nitrite ion measurement by Griess assay, (**B**) arginase activity assay, and (**C**) measurement of IL-1, IL-6, CCL2, and OPN concentrations by ELISA. Calcitriol lowers the concentration of nitrite ion, OPN, IL-1, and IL-6 in M1 BMDMs. Calcitriol enhances arginase activities for M1 and M2 BMDMs and secretion of CCL2 in M2 BMDMs. ELISA assays and measurement of arginase activity and nitrite ions concentration (Griess assay) were performed according to the manufacturers’ protocols. Data presentation: mean with standard deviation. Number of independent repetitions = 3 (**C**) or 4 (**A**,**B**) (BMDMs cultures generated from three or four mice). Statistical analysis: Sidak’s or Dunn’s multiple comparisons test. * *p* < 0.05 as compared to M0, ^ *p* < 0.05 as compared to M0 + cal or as indicated.

**Figure 5 cancers-12-03485-f005:**
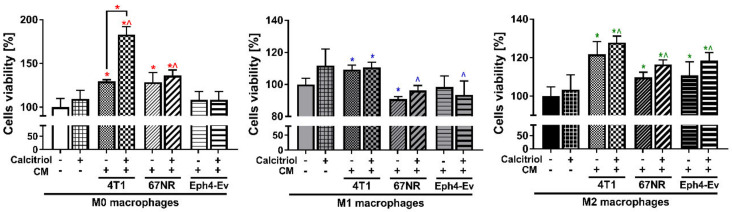
Proliferation of various BMDMs classes stimulated with conditioned media (CM), with or without calcitriol. 4T1 CM have the greatest impact on macrophage proliferation, regardless of the BMDMs class. All CM stimulate the proliferation of M2 BMDMs. The effect of CM from 4T1, 67NR, and Eph4-Ev cultures with or without calcitriol on the proliferation of polarized BMDMs was estimated by the SRB (Sulforhodamine B) test. Briefly, 80% TCA acid was applied on the wells. After 1 h, the wells were rinsed 5 times with distilled water and 0.1% sulforhodamine B solution was added for 30-min incubation. After this time, the wells were rinsed 5 times with 1% acetic acid. The pellet was dissolved in a 10 mM TRIS solution and absorbance was measured. Absorbance results from BMDMs of individual classes (M0, M1, M2) treated with CM in combination with or without calcitriol were referred to individual BMDMs polarized without the addition of CM and calcitriol (control, 100%). Data presentation: mean with standard deviation. Number of independent repetitions = 3 (BMDM cultures generated from three mice). Statistical analysis: Sidak’s or Dunn’s multiple comparisons test. * (red) *p* < 0.05 as compared to M0 or * (blue) *p* < 0.05 to M1 or * (green) *p* < 0.05 to M2 not treated with cal and any CM, ^ (red) *p* < 0.05 as compared to M0 + cal or ^ (blue) *p* < 0.05 to M1 + cal or ^ (green) *p* < 0.05 to M2 + cal not treated with any CM; or as indicated.

**Figure 6 cancers-12-03485-f006:**
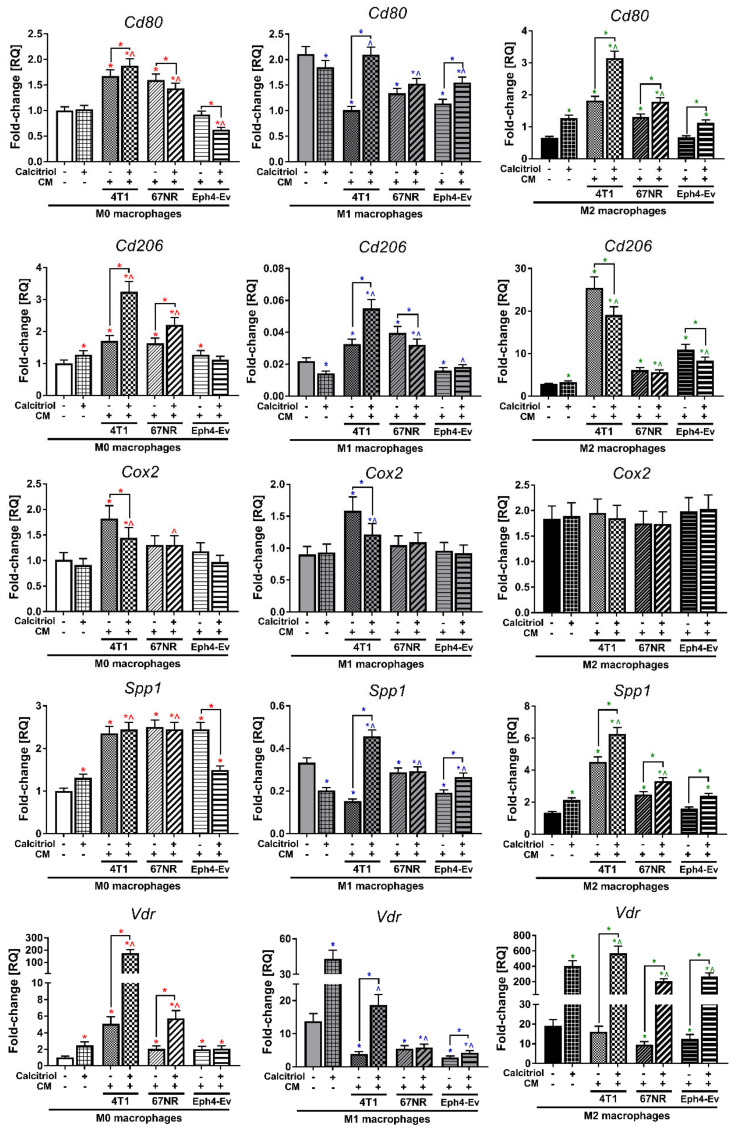
Expression of selected genes in various BMDMs classes, stimulated with CM, with or without calcitriol analyzed by real-time PCR. In general, 4T1 CM showed the highest influence on the expression of the studied genes, regardless of the BMDMs class. The greatest effect was noted for M2 BMDMs, especially after incubation with 4T1 CM +/- calcitriol. The level of gene expression is presented in relation to the results obtained for M0 BMDMs (unstimulated and untreated with calcitriol). Briefly, RNA was isolated, purified, and transcribed into cDNA. Real-time PCR was performed using Taq-Man chemistry. A single reaction was performed with 50 ng of cDNA and each sample was performed in technical triplicates. The relative quantification (RQ) level of the examined gene expression, referred to as fold change, was calculated based on the differences in the ΔΔCt values of the studied genes in relation to the control housekeeping gene *Hprt1*. Data presentation: mean with standard deviation. Number of independent repetitions = 3 (BMDMs cultures generated from three mice). Statistical analysis: Sidak’s or Dunn’s multiple comparisons test. * (red) *p* < 0.05 as compared to M0 or * (blue) *p* < 0.05 to M1 or * (green) *p* < 0.05 to M2 not treated with cal and any CM, ^ (red) *p* < 0.05 as compared to M0 + cal or ^ (blue) *p* < 0.05 to M1 + cal or ^ (green) *p* < 0.05 to M2 + cal not treated with any CM; or as indicated.

**Figure 7 cancers-12-03485-f007:**
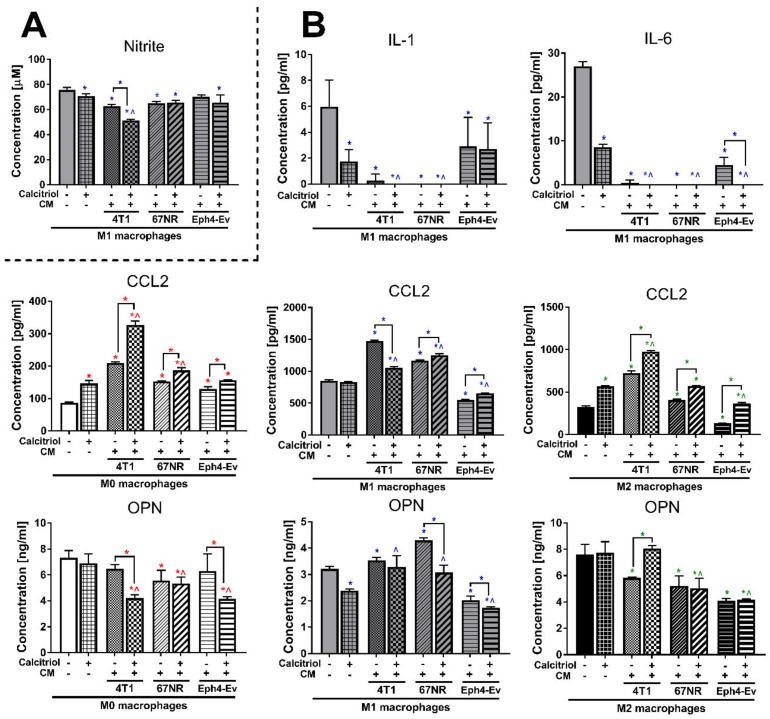
Determination of the effect of CM and calcitriol on the level of expression of markers describing individual BMDMs classes: (**A**) nitrite ion measurement by Griess assay and (**B**) measurement of IL-1, IL-6, CCL2, and OPN concentrations by ELISA. 4T1 CM +/- calcitriol decreased the concentration of nitrite, IL-1, IL-6, and CCL2 ions for M1 BMDMs and increased the concentration of CCL2 for M0 and M2 BMDMs and OPN for M2 BMDMs. ELISA assays and measurement of arginase activity and nitrite ions concentration (Griess assay) were performed according to the manufacturers’ protocols. Data presentation: mean with standard deviation. Number of independent repetitions = 3 (**B**) or 4 (**A**) (BMDM cultures generated from three or four mice). Statistical analysis: Sidak’s or Dunn’s multiple comparisons test. * (red) *p* < 0.05 as compared to M0 or * (blue) *p* < 0.05 to M1 or * (green) *p* < 0.05 to M2 not treated with cal and any CM, ^ (red) *p* < 0.05 as compared to M0 + cal or ^ (blue) *p* < 0.05 to M1 + cal or ^ (green) *p* < 0.05 to M2 + cal not treated with any CM; or as indicated.

**Figure 8 cancers-12-03485-f008:**
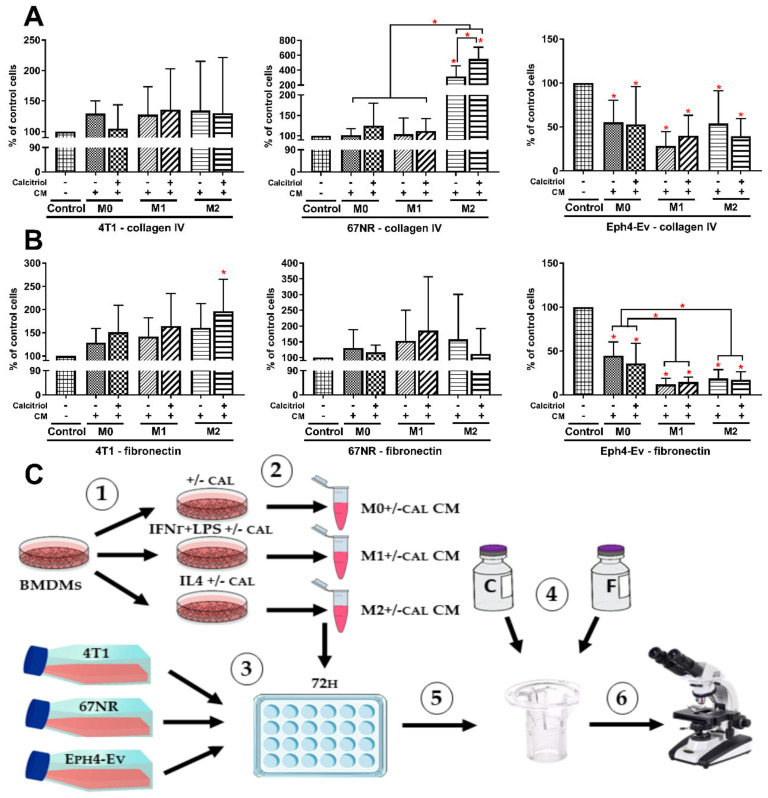
Effect of BMDMs of individual classes, differentiated with or without calcitriol, on the migration properties of normal and cancer cells: migration through (**A**) collagen type IV and (**B**) fibronectin. Both M2 BMDMs differentiated with and without calcitriol stimulated migration of 67NR cells through collagen IV, while M2+cal BMDMs increased 4T1 migration by fibronectin. The migration assay was repeated 4 times for each cell line. Statistical analysis: Sidak’s or Dunn’s multiple comparisons test. * *p* < 0.05 as compared to control or as indicated. The control cells were cells not treated with any BMDMs CM but only with serum-free DMEM (1:1 ratio). (**C**) The scheme of the experiment: (**1**) BMDMs were differentiated for 48h into different classes, with or without calcitriol. (**2**) Differentiated BMDMs were starved for 24 h in serum-free medium to generate conditioned media (CM). (**3**) Normal epithelial Eph4-Ev and cancer 4T1 and 67NR cells were seeded at the appropriate density in a 24-well plate and incubated with individual BMDMs CM in a 1:1 ratio (fresh medium/CM) for 72 h. (**4**) Membrane inserts (8 µm pore size) were coated overnight with a collagen type IV or fibronectin solution. (**5**) After incubation with BMDMs CM, cells were detached and seeded onto inserts (upper chamber). Migration was carried out for 8h in an incubator in the presence of FBS as a chemoattractant. (**6**) Cells that migrated to the other side of the insert membrane were fixed, stained, and counted under a light microscope.

**Figure 9 cancers-12-03485-f009:**
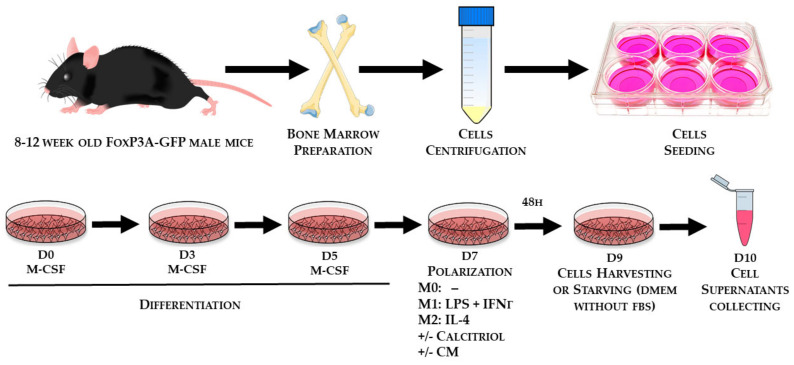
Scheme illustrating the isolation of bone marrow progenitor cells and their differentiation into macrophages of the respective classes.

**Table 1 cancers-12-03485-t001:** Production of cytokines and chemokines in the cell culture supernatants of BMDMs polarized in the presence or absence of calcitriol.

Protein	M0	M0 + cal	M1	M1 + cal	M2	M2 + cal
G-CSF	1.36%	0.81%	0.04%	1.45%	3.10% *#	3.61% *^#
GM-CSF	0.22%	0.04%	0.00%	0.00%	4.88% *#	3.67% *^#
IFN-γ	0.00%	0.00%	0.02%	1.00% *^&	0.00%	0.00% #
IL-1RA	20.50%	13.34%	79.44% *	104.76% *^&	55.42% *#	49.91% *^#
IL-10	0.00%	0.00%	0.00%	0.00%	1.32% *#	1.09% *^#
IL-27	0.00%	0.00%	0.00%	0.00%	4.56% *#	6.31% *^#
CXCL10	71.66%	45.75% *	100.04% *	90.25%	121.41% *	123.68% *^
CXCL11	1.92%	1.25%	1.08% *	2.41% ^&	0.00% *#	0.00% *^#
CXCL1	69.48%	68.10%	16.13% *	20.79% *^	124.39% *#	143.05% *^#&
CCL12	0.00%	0.00%	0.00%	0.00%	5.44% *#	3.76% *^#
CXCL9	2.00%	1.07%	13.80% *	12.27% *^	6.89% *#	4.01%
CCL3	101.63%	88.85%	1.82% *	5.72% *^	100.13% #	77.59% *#&
CCL4	79.07%	72.03%	0.49% *	3.41% *^	69.20% #	66.37% #
CXCL2	110.57%	104.17%	49.29% *	52.74% *^	83.07% *#	67.31% *^
CCL5	136.22%	126.49%	120.81%	105.06% *^	121.55%	110.80% *
CXCL12	6.45%	4.79%	4.78%	5.40%	3.46% *	3.44% *
TIMP-1	2.11%	1.55%	1.52%	2.97%	13.64% *#	17.58% *^#
TNF-α	10.82%	13.76%	109.60% *	115.36% *^	46.21% *#	48.80% *^#

The macrophages M1 and M2 show a different profile of secreted chemokines and cytokines; calcitriol reduces the level of CXCL10 in M0 BMDMs and CCL3 in M2 BMDMs and stimulates the production of IFN-γ, IL-1RA, and CXCL11 in M1 BMDMs. Briefly, samples and membranes were incubated with the detection antibody cocktail overnight. The next day, the secondary antibody was applied and chemiluminescence was detected. The expression value of the tested proteins is presented as a percentage, in relation to the positive control placed on the membrane by the producer (100% value). Standard deviations are listed in the extended Appendix A. Data presentation: mean. Number of independent repetitions = 4 (BMDMs cultures generated from four mice). Statistical analysis: Sidak’s or Dunn’s multiple comparisons test. * *p* < 0.05 as compared to M0, ^ *p* < 0.05 as compared to M0 + cal, # *p* < 0.05 as compared between M1 and M2 or between M1 + cal and M2 + cal, & *p* < 0.05 as compared between M1 and M1 + cal or between M2 and M2 + cal.

**Table 2 cancers-12-03485-t002:** Mixes of antibodies conjugated with fluorochromes used in the cytometric analysis.

Mix 1		
Antibody	Host Species	Manufacturer
F4/80- BV421	Rat	BD Biosciences
CD11b-APC	Rat	BD Biosciences
CD54-FITC	Hamster	BD Biosciences
MHC II-PerCP-Cy5.5	Rat	BioLegend
CD204-BV650	Rat	BioLegend
CD80-PeCy7	Hamster	BioLegend
CD44-BV510	Rat	BioLegend
CD163-PE	Rat	Thermo Fisher
**Mix 2**		
F4/80-BV421	Rat	BD Biosciences
CD36-PE	Mouse	BD Biosciences
CD86-FITC	Rat	BD Biosciences

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
