# Peer review of "Calcitriol in the Presence of Conditioned Media from Metastatic Breast Cancer Cells Enhances Ex Vivo Polarization of M2 Alternative Murine Bone Marrow-Derived Macrophages"

_cancers, 2020, doi:10.3390/cancers12113485_

Round 1

Reviewer 1 Report

I dont have more comments.

Reviewer 2 Report

Thanks to the authors for clarifying doubts and following the recommendations. I look forward to seeing their future work in the same line.  

Reviewer 3 Report

authors provide fair responses and justification in response to the previous reviews.

Legends are not yet satisfactory.

Legends must include a stand-alone description of the method together with indications on the number of performed experiments. Legends must also state information on statistics.

Author Response

Reviewer 3:

authors provide fair responses and justification in response to the previous reviews.

Legends are not yet satisfactory.

Legends must include a stand-alone description of the method together with indications on the number of performed experiments. Legends must also state information on statistics.

Dear Reviewer,

We have modified and extended the descriptions to all figures with a brief description of the method (highlighted in yellow). We carefully checked all the figures to make sure that they all contain information about the performed statistical test with the number of experimental repetitions.

Figure 1. The effect of calcitriol on gene expression in individual classes of BMDMs analyzed by Real-Time PCR. Calcitriol lowers Cd80 and Spp1 expression for M1 BMDMs, increases Cd80 expression for M2 BMDMs and Cd206 and Spp1 for M0 and M2 BMDMs, and increases Vdr expression in all BMDMs classes. The level of gene expression is presented in relation to the results obtained for M0 BMDMs (unstimulated and untreated with calcitriol). Briefly, RNA has been isolated, purified, and transcribed into cDNA. Real-time PCR was performed using Taq-Man chemistry. A single reaction was performed with 50 ng of cDNA and each sample was performed in technical triplicates. The relative quantification (RQ) level of examined gene expression, referred to as fold change, was calculated based on the differences in ΔΔCt values of the studied genes in relation to the control housekeeping gene Hprt1. Data presentation: mean with standard deviation. Number of independent repetitions = 3 (BMDMs cultures generated from three mice). Statistical analysis: Sidak’s or Dunn’s multiple comparisons test. *p<0.05 as compared to M0, ^p<0.05 as compared to M0 + cal or as indicated.

Figure 2. The effect of calcitriol on the expression of surface markers in individual classes of BMDMs analyzed by FACS: (A) Pan-macrophage markers, (B) M1 markers, and (C) M2 markers. Calcitriol enhances CD11b and MHC II expression for M1 BMDMs, lowers CD80 expression and increases CD36 expression for M2 BMDMs. Briefly, BMDMs were detached with a non-enzymatic solution. 0.1 x 105 cells incubated with anti-mouse CD16/32 for blocking non-specific binding of immunoglobulin to the Fc receptors. Then individual antibodies or isotype controls were applied and Fixable Viability Dye eFluor 780 for viability control. The results are presented as percentage of positive cells expressing the examined molecule and as Median Fluorescence Intensity (MFI). The gating strategy and images of representative histograms for each BMDMs class are provided in Supplementary Figure S2. Data presentation: mean with standard deviation. Number of independent repetitions = 3 (BMDMs cultures generated from three mice). Statistical analysis: Sidak’s or Dunn’s multiple comparisons test. *p<0.05 as compared to M0, ^p<0.05 as compared to M0 + cal or as indicated.

Figure 3. The influence of calcitriol on the production of cytokines and chemokines in the supernatants of BMDMs cultures analyzed by Mouse Cytokine Array Panel A assay. The macrophages M1 and M2 show a different profile of secreted chemokines and cytokines; calcitriol reduces the level of CXCL10 in M0 BMDMs and CCL3 in M2 BMDMs and stimulates the production of IFN-γ, IL-1RA and CXCL11 in M1 BMDMs. Briefly, samples and membranes were incubated with the detection antibody cocktail overnight. The next day, the secondary antibody was applied and chemiluminescence was detected. The level of expression is presented as a heat map based on the results presented in Table 1. Data presentation: mean. Number of independent repetitions = 4 (BMDMs cultures generated from four mice).

Table 1. Production of cytokines and chemokines in the cell culture supernatants of BMDMs polarized in the presence or absence of calcitriol. The macrophages M1 and M2 show a different profile of secreted chemokines and cytokines; calcitriol reduces the level of CXCL10 in M0 BMDMs and CCL3 in M2 BMDMs and stimulates the production of IFN-γ, IL-1RA and CXCL11 in M1 BMDMs. Briefly, samples and membranes were incubated with the detection antibody cocktail overnight. The next day, the secondary antibody was applied and chemiluminescence was detected. The expression value of the tested proteins is presented as a percentage, in relation to the positive control placed on the membrane by the producer (100% value). Standard deviations are listed in the extended Supplementary Table S1. Data presentation: mean. Number of independent repetitions = 4 (BMDMs cultures generated from four mice). Statistical analysis: Sidak’s or Dunn’s multiple comparisons test. *p<0.05 as compared to M0, ^p<0.05 as compared to M0 + cal, #p<0.05 as compared between M1 and M2 or between M1 + cal and M2 + cal, &p<0.05 as compared between M1 and M1 + cal or between M2 and M2 + cal.

Figure 4. Determination of the effect of calcitriol on the level of expression of markers describing individual BMDMs classes: (A) nitrite ion measurement by Griess assay, (B) arginase activity assay, and (C) measurement of IL-1, IL-6, CCL2, and OPN concentrations by ELISA. Calcitriol lowers the concentration of nitrite ion, OPN, IL-1 and IL-6 in M1 BMDMs. Calcitriol enhances arginase activities for M1 and M2 BMDMs and secretion of CCL2 in M2 BMDMs. ELISA assays and measurement of arginase activity and nitrite ions concentration (Griess assay) were performed according to the manufacturers' protocols. Data presentation: mean with standard deviation. Number of independent repetitions = 3 (C) or 4 (A and B) (BMDMs cultures generated from three or four mice). Statistical analysis: Sidak’s or Dunn’s multiple comparisons test. *p<0.05 as compared to M0, ^p<0.05 as compared to M0 + cal or as indicated.

Figure 5. Proliferation of various BMDMs classes stimulated with CM, with or without calcitriol. 4T1 CM has the greatest impact on macrophage proliferation, regardless of the BMDMs class. All CMs stimulate the proliferation of M2 BMDMs. The effect of CM from 4T1, 67NR, and Eph4-Ev cultures with or without calcitriol on the proliferation of polarized BMDMs was estimated by the SRB test. Briefly, 80% TCA acid was applied on wells. After 1 h, the wells were rinsed 5 times with distilled water and 0.1% sulphorhodamine B solution was added for 30 min incubation. After this time, the wells were rinsed 5 times with 1% acetic acid. The pellet was dissolved in a 10 mM TRIS solution and absorbance was measured. Absorbance results from BMDMs of individual classes (M0, M1, M2) treated with CM in combination with or without calcitriol were referred to individual BMDMs polarized without the addition of CM and calcitriol (control, 100%). Data presentation: mean with standard deviation. Number of independent repetitions = 3 (BMDM cultures generated from three mice). Statistical analysis: Sidak’s or Dunn’s multiple comparisons test. *p<0.05 as compared to M0 or *p<0.05 to M1 or *p<0.05 to M2 not treated with cal and any CM, ^p<0.05 as compared to M0 + cal or ^p<0.05 to M1 + cal or ^p<0.05 to M2 + cal not treated with any CM; or as indicated.

Figure 6. Expression of selected genes in various BMDMs classes, stimulated with CM, with or without calcitriol analyzed by Real-Time PCR. In general, 4T1 CM showed the highest influence on the expression of the studied genes, regardless of the BMDMs class. The greatest effect was noted for M2 BMDMs, especially after incubation with 4T1 CM +/- calcitriol. The level of gene expression is presented in relation to the results obtained for M0 BMDMs (unstimulated and untreated with calcitriol). Briefly, RNA has been isolated, purified, and transcribed into cDNA. Real-time PCR was performed using Taq-Man chemistry. A single reaction was performed with 50 ng of cDNA and each sample was performed in technical triplicates. The relative quantification (RQ) level of examined gene expression, referred to as fold change, was calculated based on the differences in the ΔΔCt values of the studied genes in relation to the control housekeeping gene Hprt1. Data presentation: mean with standard deviation. Number of independent repetitions = 3 (BMDMs cultures generated from three mice). Statistical analysis: Sidak’s or Dunn’s multiple comparisons test. *p<0.05 as compared to M0 or *p<0.05 to M1 or *p<0.05 to M2 not treated with cal and any CM, ^p<0.05 as compared to M0 + cal or ^p<0.05 to M1 + cal or ^p<0.05 to M2 + cal not treated with any CM; or as indicated.

Figure 7. Determination of the effect of CM and calcitriol on the level of expression of markers describing individual BMDMs classes: (A) nitrite ion measurement by Griess assay and (B) measurement of IL-1, IL-6, CCL2, and OPN concentrations by ELISA. 4T1 CM +/- calcitriol decreased the concentration of nitrite, IL-1, IL-6 and CCL2 ions for M1 BMDMs and increased the concentration of CCL2 for M0 and M2 BMDMs and OPN for M2 BMDMs. ELISA assays and measurement of arginase activity and nitrite ions concentration (Griess assay) were performed according to the manufacturers' protocols. Data presentation: mean with standard deviation. Number of independent repetitions = 3 (B) or 4 (A) (BMDM cultures generated from three or four mice). Statistical analysis: Sidak’s or Dunn’s multiple comparisons test. *p<0.05 as compared to M0 or *p<0.05 to M1 or *p<0.05 to M2 not treated with cal and any CM, ^p<0.05 as compared to M0 + cal or ^p<0.05 to M1 + cal or ^p<0.05 to M2 + cal not treated with any CM; or as indicated.

Figure 8. Effect of BMDMs of individual classes, differentiated with or without calcitriol, on the migration properties of normal and cancer cells: migration through (A) collagen type IV and (B) fibronectin. Both M2 BMDMs differentiated with and without calcitriol stimulated migration of 67NR cells through collagen IV, while M2+cal BMDMs increased 4T1 migration by fibronectin. The migration assay was repeated 4-times for each cell line. Statistical analysis: Sidak’s or Dunn’s multiple comparisons test. *p<0.05 as compared to Control or as indicated. The control cells were cells not treated with any BMDMs CM but only with serum-free DMEM (1:1 ratio). (C) The scheme of the experiment: (1) BMDMs were differentiated 48h into different classes, with or without calcitriol. (2) Differentiated BMDMs were starved 24 h in serum-free medium to generate conditioned media (CM). (3) Normal epithelial Eph4-Ev and cancer 4T1 and 67NR cells were seeded at the appropriate density in a 24-well plate and incubated with individual BMDMs CM in a 1:1 ratio (fresh medium : CM) for 72 h. (4) Membrane inserts (8 µm pore size) were coated overnight with a collagen type IV or fibronectin solution. (5) After incubation with BMDMs CM cells were detached and seeded onto inserts (upper chamber). Migration was carried out for 8h in an incubator in the presence of FBS as a chemoattractant. (6) Cells that migrated to the other side of the insert membrane were fixed, stained and counted under a light microscope.

This manuscript is a resubmission of an earlier submission. The following is a list of the peer review reports and author responses from that submission.

Round 1

Reviewer 1 Report

In this study the authors demonstrate vitamin D supplementation may not always be beneficial, especially in relation to cancers causing excessive, pathological activation of the immune system. The studies are nicely executed, and most of the findings are straight forward. While the concept of the work is very interesting and informative, there are a few major concerns that the authors need to address in their study. 

  1. Please provide significance of the research in more details.
  2. Please provide more representative images for immunofluorescence microscopy images in figures 1C
  3. Please show FACS profile for Figure 3
  4. Please improve discussion section
  5. Please cite the following articles in your manuscript and include in reference section
    1. Spleen Tyrosine Kinase–Mediated Autophagy Is Required for Epithelial–Mesenchymal Plasticity and Metastasis in Breast Cancer.
    2. Inhibition of pyruvate carboxylase by 1α, 25-dihydroxyvitamin D promotes oxidative stress in early breast cancer progression

Reviewer 2 Report

This is a descriptive pre-clinical study examining the role of calcitriol in modifying macrophage polarization ex vivo utilizing bone marrow-derived macrophages (BMDMs). The authors conclude that calcitriol promotes the polarization of BMDMs toward the M2 phenotype ex vivo which is supported by evaluation of classic M0, M1, and M2 macrophage markers. Additionally, the authors conclude that 4T1 (metastatic) conditioned media (CM) had greater potential to affect gene and protein expression in BMDMs compared to 67NR (non-metastatic) or Eph4-Ev (normal) CM, with the greatest effects seen in M2 macrophages which increased their differentiation and properties characteristic of alternative macrophages, which could be explained by differential expression of the vitamin D receptor among M0, M1, and M2 macrophages. These results demonstrate potential therapeutic considerations, with respect to supplementation with vitamin D or calcitriol derivatives, which may not always be beneficial due to the immunosuppressive properties of vitamin D and may results in cancer progression. However, novelty with respect to the characterization of BMDMs polarization is minimal and while the effects of calcitriol on BMDMs polarization may be novel, the authors have previously published several studies describing the effects of calcitriol in vitro and in vivo which seem to crossover with the results in this study.

The following highlight areas of concern or where improvement is needed.

  • The introduction and discussion sections of the manuscript read as a review article and it is unclear the overall goal of the study as there are many topics included in both sections, some of which are not pertinent to this study (i.e. discussion of age-related effects of calcitriol in vivo – no results presented here to demonstrate this).
  • The figure legends provide a descriptive title to detail what experiments were performed in each figure, but the reader is left to decipher the overall conclusion of each figure – needs to be modified to improve clarity of the manuscript.
  • The supplemental data figures are not included until the discussion and should be included in the results section, if relevant to this manuscript.
  • In Figure 1, were the M1 and M2 BMDMs stained with a specific M1 or M2 marker? Based on the images provided, it is difficult to truly tell the difference between the M1 or M2 phenotype as both images look quite similar. We also question the novelty and relevance of Figure 1 as this is well known throughout the field of deriving macrophages from murine bone marrow.
  • The title of the article concludes that calcitriol in the presence of CM from metastatic breast cancer cells polarizes BMDMs toward the M2 phenotype but based on how the results are presented, it appears that the BMDMs were already skewed towards M0, M1, or M2 phenotype and then co-cultured with calcitriol and CM. Did the authors perform the experiment with M0 BMDMs and evaluate differences in macrophage polarization? If the authors are claiming that calcitriol affects macrophage polarization, it would be important to determine if the effects of calcitriol are truly on polarization or if they enhance the M2 phenotype in the tumor microenvironment.
  • The effects of macrophage polarization and calcitriol are well described in the literature, while the authors provide descriptive evidence to attempt to relate calcitriol to macrophage polarization, they do not provide further in vitro or in vivo experiments to strengthen their argument that calcitriol drives the M2 phenotype which severely limits this manuscript to add to the field.
  • To heighten novelty, several experiments should be undertaken and may include:
    1. Culture M0 BMDMs with calcitriol and 4T1 CM and evaluate the effects of calcitriol on macrophage polarization – do they truly drive M2 macrophage polarization?
    2. Treat tumor bearing mice with calcitriol (and with other therapeutics) to evaluate the effects of tumor progression and metastatic as well as characterization of the tumor microenvironment
    3. Knockdown or knockout the Vitamin D receptor in M2 macrophages and evaluate the effects with or without calcitriol plus CM
    4. If calcitriol is synergistic with pro-tumor cell and M2 macrophage properties, perform co-culture assays with tumor cells and M2 macrophages with or without calcitriol (growth curves, migration/invasion assays, sphere formation, etc.)
    5. Are the correlations between vitamin supplements and breast cancer patients (poor survival, increased metastasis, etc.)?

Reviewer 3 Report

Anisiewicz and collagues show in this paper that calcitriol can promote the ex vivo polarization of murine bone marrow-derived macrophages (BMDMs) to the M2 phenotype. Moreover, in order to partially mimic the TME conditions, they generated conditioned media (CM) from 4T1 metastatic cells, nonmetastatic 67NR cells, and normal epithelial Eph4-Ev cells and examined how CM alone and in combination with calcitriol affected some properties of BMDMs. 4T1 CM showed a higher effect than 67NR and Eph4-Ev, with the greatest effect observed on M2 macrophages which increased their differentiation and properties, so the authors suggest that vitamin D supplementation may not be beneficial in the course of neoplasic diseases.

Overall, the paper is nicely written, the introduction and the results are composed in a competent style. The authors performed a rigorous analysis of the data obtained from the experiments and the conclusions are coherent.

I have only a few comments that should be corrected or considered before publication:

1-What was the point of using FoxP3A-GFP mice if they did not analyze Treg cells?

2- Calcitriol (100 nM). Why this dose and no another?

3- The authors used * and Λ to point out statistically significant differences between several conditions. In some figures it was a little bit confusing, so add a new symbol would be helpful.

4- In the line 402, IL-6 has been written twice.

5- 4T1 CM had the greater potential to affect gene and protein expression in BMDMs than 67NR CM and Eph4-Ev CM, with the greatest effect seen in M2 macrophages. The authors suggest this phenomenon could be explained by the differential expression of the vitamin D receptor in individual classes of macrophages. A nice control will be the use of specific antagonist of VDR in the cultures.

Round 2

Reviewer 1 Report

The authors have addressed all the comments.

Reviewer 2 Report

This is a descriptive pre-clinical study looking at the role of calcitriol to enhance macrophage polarization ex vivo utilizing bone marrow-derived macrophages (BMDMs). The results presented in this study demonstrate potential therapeutic considerations, with respect to supplementation with vitamin D or calcitriol derivatives, which may not always be beneficial due to the immunosuppressive properties of vitamin D and may results in cancer progression. We as reviewers appreciate the changes the authors made within the introduction and discussion sections of the manuscript to better address the goals of this study. However, the authors fail to strengthen the novelty of this study as much of the data presented here is not novel with respect to BMDM polarization. While the effects of calcitriol on BMDMs polarization may be novel, the authors have previously published several studies describing the effects of calcitriol in vitro and in vivo which seem to crossover with the results in this study. Additionally, the authors indicate in their rebuttal that several key functional and mechanistic studies are ongoing which should be included to strengthen their claims in this manuscript and add to novelty.